# Protection against lethal canine distemper virus infection by a dual epitope-targeting synthetic antibody

Melanie Scherer [1,2,10,12], Nadia Djabeur[2,3,12], Oliver Siering[4,12], Jean-Marc Jeckelmann [3], Marianne Wyss[1], Marina Cresci[5], Morgane Di Palma Subran[5,11], Rainer Riedl [6], Patrick Chames [5], Christian K. Pfaller [4,7,8], Bevan Sawatsky [4,13] ✉, Dimitrios Fotiadis [3,9,13] ✉ & Philippe Plattet [1,9,13] ✉

Despite vaccine availability, the morbilliviruses measles virus and canine distemper virus (CDV) are still causing major health impairments in human and animal populations. Here, we identified two potent, neutralizing single domain antibodies directed against the tetrameric receptor binding (H) protein of CDV. Structural analyses spotlighted two vulnerable sites within the H protein. While the first overlaps with the receptor binding site, the second encompasses amino acid residues of two protomers located at the distal dimeric head interface, which supports distinct mechanisms of neutralization. Upon application of an engineered tetravalent and biparatopic antibody, ferrets were protected at a remarkably low antibody dose (1 mg/kg) administered intraperitoneally on days 3 and 7 post-exposure of a lethal CDV challenge. Collectively, this study spotlights the power of integrating multiple mechanisms of neutralization in a single format and provides a roadmap to design next-generation therapeutics against morbilliviral infections as well as other infectious pathogens.

Despite the availability of effective and safe vaccines, morbilliviruses such as measles virus (MeV) and canine distemper virus (CDV) are still causing important global health and economic impairments[1]. In 2019, MeV caused more than 200,000 deaths in the human population[2], whereas CDV continues to induce severe outbreaks in wild animal species as well as in dogs in countries where vaccination campaigns remain suboptimal. Furthermore, CDV has been shown to cross the species barrier, posing a significant threat to certain endangered species[3,4]. Although MeV and CDV are both neurotropic pathogens, CDV exhibits the highest incidence with up to 30% of infected dogs developing neurological symptoms[3,5]. Morbilliviruses belong to the family *Paramyxoviridae*, which also includes the emerging zoonotic henipaviruses, Nipah virus and Hendra virus, that have exceptionally high fatality rates[6]. While antivirals would be highly beneficial for

[1]Division of Neurological Sciences, Vetsuisse Faculty, University of Bern, Bern, Switzerland. [2]Graduate School for Cellular and Biomedical Sciences, University of Bern, Bern, Switzerland. [3]Institute of Biochemistry and Molecular Medicine, Medical Faculty, University of Bern, Bern, Switzerland. [4]Division of Veterinary Medicine, Paul-Ehrlich-Institute, Langen, Germany. [5]Aix-Marseille Université, CNRS, INSERM, Institute Paoli-Calmettes, CRCM, Marseille, France. [6]Competence Center for Drug Discovery, ZHAW Zurich University of Applied Sciences, Institute of Chemistry and Biotechnology, Wädenswil, Switzerland. [7]Mayo Clinic Graduate School of Biomedical Sciences, Virology and Gene Therapy Track, Rochester, MN, USA. [8]Department of Molecular Medicine, Mayo Clinic, Rochester, MN, USA. [9]Multidisciplinary Center for Infectious Diseases (MCID), University of Bern, Bern, Switzerland. [10]Present address: Institute of Medical Microbiology, University of Zurich, Zürich, Switzerland. [11]Present address: Division of Neurological Sciences, Vetsuisse Faculty, University of Bern, Bern, Switzerland. [12]These authors contributed equally: Melanie Scherer, Nadia Djabeur, Oliver Siering. [13]These authors jointly supervised this work: Bevan Sawatsky, Dimitrios Fotiadis, Philippe Plattet. ✉e-mail: bevan.sawatsky@pei.de; dimitrios.fotiadis@unibe.ch; philippe.plattet@unibe.ch

better disease management of these pathogens, no anti-paramyxoviral drug has currently been authorized by the Food and Drug Administration (FDA).

Morbilliviruses infect lymphoid, epithelial and neurological cells leading to associated clinical signs[7,8]. To enter a target cell, they usually rely on two interacting envelope glycoproteins: a receptor-binding tetrameric (dimer-of-dimers) attachment (H) protein and a trimeric fusion (F) protein[9]. The CDV H protein classifies as type II transmembrane protein, consisting of a N-terminal cytosolic tail, a transmembrane domain and a large C-terminal ectodomain that can be further divided into three main regions: a membrane-proximal stalk (containing F-interaction and activation sites), an intermediate neck segment and a membrane-distal cubic head domain that encompasses the receptor binding sites (RBS)[10,11]. Recent cryogenic-electron microscopy (cryo-EM) structural analysis revealed an inherent asymmetric architecture of the tetrameric CDV H ectodomain, primarily attributed to the presence of the neck domain. Indeed, the neck induces an almost 90° bend, positioning the head domains - organized in two non-contacting dimeric units - on one side of a straight, helical tetrameric stalk domain[12]. Remarkably, this CDV H architecture may allow the binding of trimeric prefusion F complexes to the H stalk domain without (i) sterically clashing with the H-heads or (ii) preventing their receptor binding function. Such putative "pre-receptor-bound" native conformational state is consistent with previous mechanistic studies suggesting that H and F glycoproteins assemble as hetero-oligomeric complexes on the viral envelope prior to receptor binding[13].

The morbilliviral cell entry process is assumed to be highly dynamic. Upon receptor binding, e.g., signaling lymphocyte activation molecule [CD150/SLAM] in immune cells[14-16] or nectin-4 in epithelial cells[17,18], the H heads may rearrange to unleash the inherent F-triggering activity of the H stalk[19-25]. Trimeric F proteins may then undergo a series of structural rearrangements from an initial metastable prefusion conformation to a final and highly stable postfusion state, presumably via the transient folding of an intermediate structure referred to as the prehairpin conformation[9,26]. Such drastic conformational changes lead to the merging of the viral envelope with the host cell plasma membrane and the formation of a fusion pore. Apart from virus-to-cell fusion, H and F expression in infected cells further mediate cell-to-cell fusion, which eventually (and depending on the cell type) results in the formation of giant multi-nucleated cells (also called syncytia): a hallmark of paramyxovirus infection[9,27,28]. Noteworthy, MeV and CDV exhibit only one serotype[29-31], highlighting their incapacity to efficiently escape the immune response if potently elicited. Such restriction in conformational plasticity may translate into valuable clinical advantages for the development of future antiviral drugs and vaccines targeting both surface glycoproteins of these morbilliviruses.

Although several studies identified various classes of inhibitors targeting paramyxoviral proteins[32-36], no first-in-class antiviral has been approved so far. However, small molecule inhibitors interfering the viral RNA-dependent RNA polymerase showcased promising efficacies against MeV and CDV infections[37-39]. Two major families of inhibitors targeting the F protein were also identified. The first group consists of prefusion F state-stabilizing molecules, e.g., fusion inhibitory peptide (FIP: Z-D-Phe-L-Phe-Gly-OH) and 2-phenylacetanilide scaffold molecules such as the AS-48 and 3G compounds, which interact at the base of the head in the prefusion state[32,40-42]. The second group is composed of additional peptides likely binding to the prehairpin F conformation, thus blocking the F-refolding cascade necessary to induce membrane fusion[43,44]. Additionally, monoclonal antibodies (mAbs) represent another valuable class of antivirals. For instance, antibodies targeting the receptor-binding protein of henipaviruses effectively protected ferrets and African green monkeys from lethal challenges[45-49]. Several neutralizing mAbs binding to the MeV F and H proteins were also identified, with some also providing

protection in vivo. Additionally, soluble constructs of MeV cellular receptors were also shown to feature protection in animal models[50-52].

Increasing evidence demonstrates that mAbs are emerging as effective tools to fight infectious diseases. Several mAbs have received FDA approval, including: Palivizumab, which blocks respiratory syncytial virus[53]; Inmazeb, a cocktail of three mAbs targeting the Ebola virus[54] and REGN-CoV, a mixture of two mAbs that specifically target the spike protein of SARS-CoV-2[55,56]. Alternatively, single domain antibodies (also known as nanobodies® [Nbs]) are antigen-binding fragments of naturally occurring heavy-chain-only antibodies found in camelids[57], and have been shown to be effective tools in basic, applied and clinical research[58]. Nbs exhibit several major advantages as compared to conventional mAbs. Among others, they are (i) relatively small (12–15 kDa), (ii) highly stable, (iii) can be engineered into various formats, (iv) they usually exhibit high affinity to their targeted antigens, and (v) they can be produced at relatively low cost[59,60]. Recently, several very potent neutralizing Nbs against SARS-CoV-2[61], Human Immunodeficiency Virus (HIV)[62] or influenza A and B viruses[63] have been described, with some derived constructs providing promising efficacies in vivo.

Here, we report the discovery of two potent neutralizing nanobodies, Nb H7 and Nb H9, which target the receptor-binding H protein of CDV at two distinct, non-overlapping epitopes. Their mechanisms of inhibition were investigated through functional, biochemical, biophysical and cryo-EM analyzes. Guided by these mechanistic and structural insights, a highly potent tetravalent, biparatopic neutralizing antibody was designed and produced. Finally, we assessed the prophylactic and therapeutic efficacy of this engineered multidomain antibody in the CDV/ferret model system, which provides a powerful and cost-effective framework to evaluate morbillivirus therapies.

## Results

### Identification of two neutralizing Nbs binding to non-overlapping sites on the CDV H protein

To identify potent neutralizing anti-CDV H Nbs, we immunized llamas with a recombinant soluble CDV H ectodomain (solH) from the wild type A75/17-CDV strain[12]. Upon selection by phage display and subsequent screening with ELISA, we identified a panel of CDV H monovalent binders. Among those, the two Nbs H7 and H9 displayed binding against a full-length, membrane-anchored form of the receptor-binding H protein upon expression in mammalian cells, as demonstrated by immunofluorescence staining (Supplementary Fig. 1a). To measure the binding affinities of the two Nbs towards CDV H, we conducted surface plasmon resonance (SPR) analyzes with solH immobilized on the SPR chip and flowing the individual purified Nbs at various concentrations. Notably, while Nb H7 exhibited a binding affinity (measured $K_D$ values) of about 2.3 nM, Nb H9 demonstrated a binding affinity of about 5.4 nM (Supplementary Fig. 1b, c and Table 1). Furthermore, no efficient binding activity was recorded when both Nbs were tested against the MeV H protein in ELISA, which corroborates

### Table 1 | Summary of binding affinities and neutralization activities of various nanobody (Nb)-based molecules

| Binders | $K_D{}^*$ (nM) | IC$_{50}{}^\#$ (ng/ml) | IC$_{50}{}^\#$ (pM) |
|---|---|---|---|
| Nb H7 | 2.3 | 40 | 2500 |
| Nb H9 | 5.4 | 1 | 60 |
| Nb H7+Nb H9 | nd | 4 | 120 |
| H7-Fc | nd | 0.6 | 7 |
| H9-Fc | nd | 0.4 | 5 |
| H7-H9 | 1.9 | 0.02 | 0.6 |
| H7-H9-fFc | nd | 0.007 | 0.06 |

*nd* not determined, $K_D{}^*$ values were determined with the TraceDrawer software, IC$_{50}{}^\#$ values were determined with GraphPad Prism v.9.

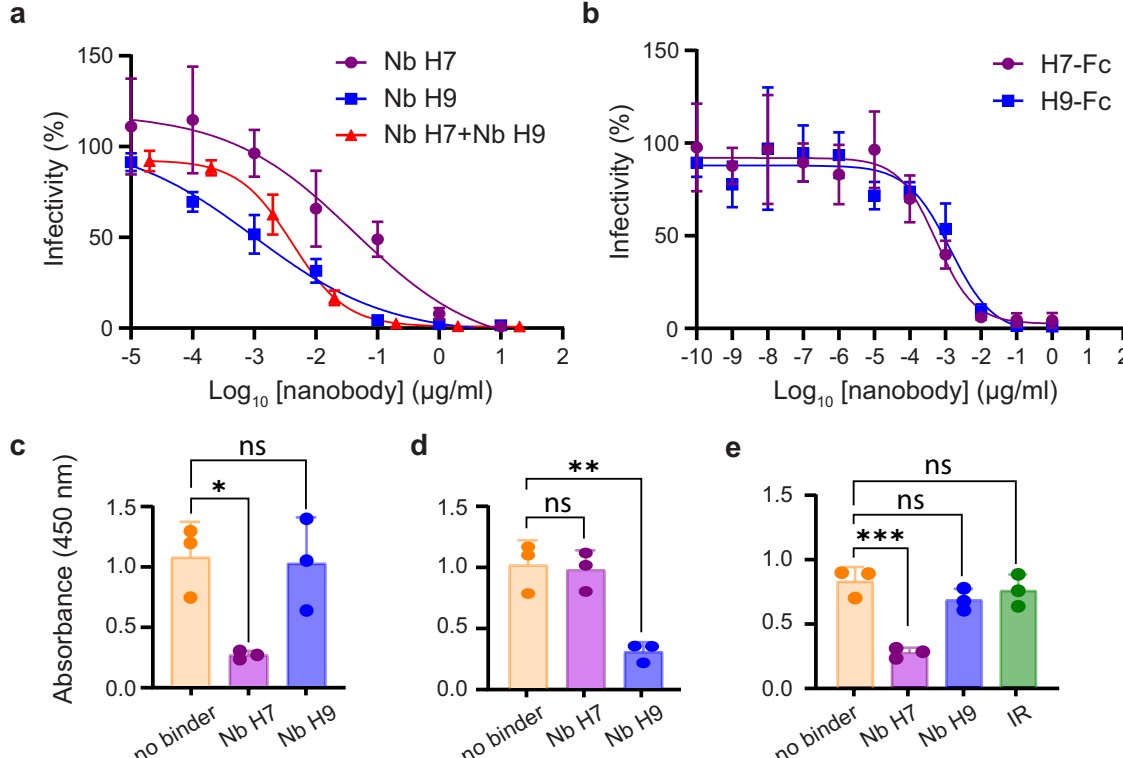

**Fig. 1 | Characterization of the anti-CDV H nanobodies H7 and H9.** Neutralization assays against CDV-A75/17[neon/nlucP]. **a** The panel depicts the relative viral infectivity in the presence of increasing concentrations of the corresponding monovalent nanobodies, as well as **b** the bivalent Nb-Fc constructs. Data are mean ± standard deviation (SD) calculated from $n = 3$ independent experiments performed in technical triplicates. **c** Competition ELISA assays. Nb H7 was immobilized on an ELISA plate, followed by exposure of TST-tagged solH in complex with Nb H7, H9, or no Nb (indicated on the x-axis). Data are mean ± SD calculated from $n = 3$ independent experiments performed in technical triplicates. **d** Same experimental setup as described in (**c**) but with Nb H9 coated in the plate. Data are mean ± SD calculated from $n = 3$ independent experiments performed in technical triplicates.

**e** Competition of the corresponding Nbs and SLAM to CDV H binding by ELISA. The recombinant solH protein was immobilized on a plate and followed by addition of Nb H7, H9, or an irrelevant (IR) Nb (indicated on the x-axis). Subsequently, plates were exposed to a soluble Fc-carrying version of SLAM and a labelled secondary antibody for binding detection. Data are mean ± SD calculated from $n = 3$ independent experiments performed in 3 technical replicates (with binders) and in 2 technical replicates (without binders). Statistical significance (**c**–**e**) was determined using a one-way ANOVA followed by Dunnett's multiple-comparison test, using GraphPad Prism v.9 ($^{*}P \leq 0.05$; $^{**}P \leq 0.01$; $^{***}P \leq 0.001$; ns: non-significant). Curves (**a**, **b**) were plotted with GraphPad Prism v.9.

with the low amino acid sequence identity between the CDV and MeV receptor binding proteins (about 35%; Supplementary Fig. 1d and Supplementary Data 5).

To investigate whether both Nbs could inhibit viral cell entry, we assessed their neutralization activity in Vero-cSLAM cells infected with the wild-type virus A75/17[neon/nlucP]. This strain was engineered to express two reporters (mNeonGreen [neon] and Nanoluciferase [nlucP]) to facilitate the recording of infected cells[36,43]. Of note, our nlucP-based viral neutralization assay, which relies on the direct recording of the luminescence emission produced in infected cells, is highly sensitive. This resulted in the determination of half maximal inhibitory concentrations (IC$_{50}$), which were specific to this assay. Both Nbs returned IC$_{50}$s of neutralization with values of about 2.5 nM and 60 pM for Nb H7 and H9, respectively (Fig. 1a and Table 1). When both nanobodies were added simultaneously at identical concentrations, we obtained IC$_{50}$ values similar to H9 alone (i.e., about 120 pM), which did not indicate any substantial additive effects (Fig. 1a, and Table 1). However, when a canine IgG Fc domain was grafted to both Nbs, and hence providing bivalent constructs, a significant improvement in neutralization potencies was recorded. Indeed, Nb H7 exhibited a 350-fold enhancement of neutralization activity (2.5 nM for Nb H7 versus 7 pM for H7-Fc), whereas Nb H9 featured a 12-fold increase (60 pM for Nb H9 versus 5 pM for H9-Fc) (Fig. 1b, and Table 1). To determine whether Nb H7 and Nb H9 were binding to similar or discrete epitopes on CDV H, we set up a competition ELISA, where Nb H7 was coated on a plate and

followed by addition of solH preincubated with either Nb H9 or Nb H7. Binding of the solH/H9 or solH/H7 complexes to Nb H7 was finally recorded using an antibody detecting solH. The reverse experiments (plates initially coated with Nb H9) were also performed. Both assays indicated that Nb H7 and Nb H9 likely targeted two distinct, non-overlapping epitopes on the CDV H protein ectodomain (Fig. 1c, d). To further validate these results, we conducted epitope binning by SPR analysis. In this set of experiments, we recorded efficient binding of Nb H9 to solH after a first injection with Nb H7 (Supplementary Fig. 1f, g). Similar results were obtained with Nb H7 after the initial binding of Nb H9 to the immobilized solH protein (Supplementary Fig. 1h, i). These data confirmed the simultaneous binding of both Nbs to the receptor-binding protein.

To better understand the molecular mechanism by which both Nbs neutralized the viral infections, we investigated whether the single domain antibodies could inhibit the interaction between the SLAM receptor and the CDV H protein by competition ELISA. Briefly, Nb H7, Nb H9 or an irrelevant (IR) Nb were supplemented to plates coated with the soluble CDV H protein. Receptor interaction was finally determined by adding a soluble SLAM-Fc molecule[64], followed by addition of an Fc-recognizing secondary antibody. Interestingly, while Nb H7 significantly blocked the interaction of SLAM with CDV H, Nb H9 did not (Fig. 1e).

Collectively, these data highlighted the successful discovery of an efficient pair of neutralizing Nbs, H7 and H9, that could bind

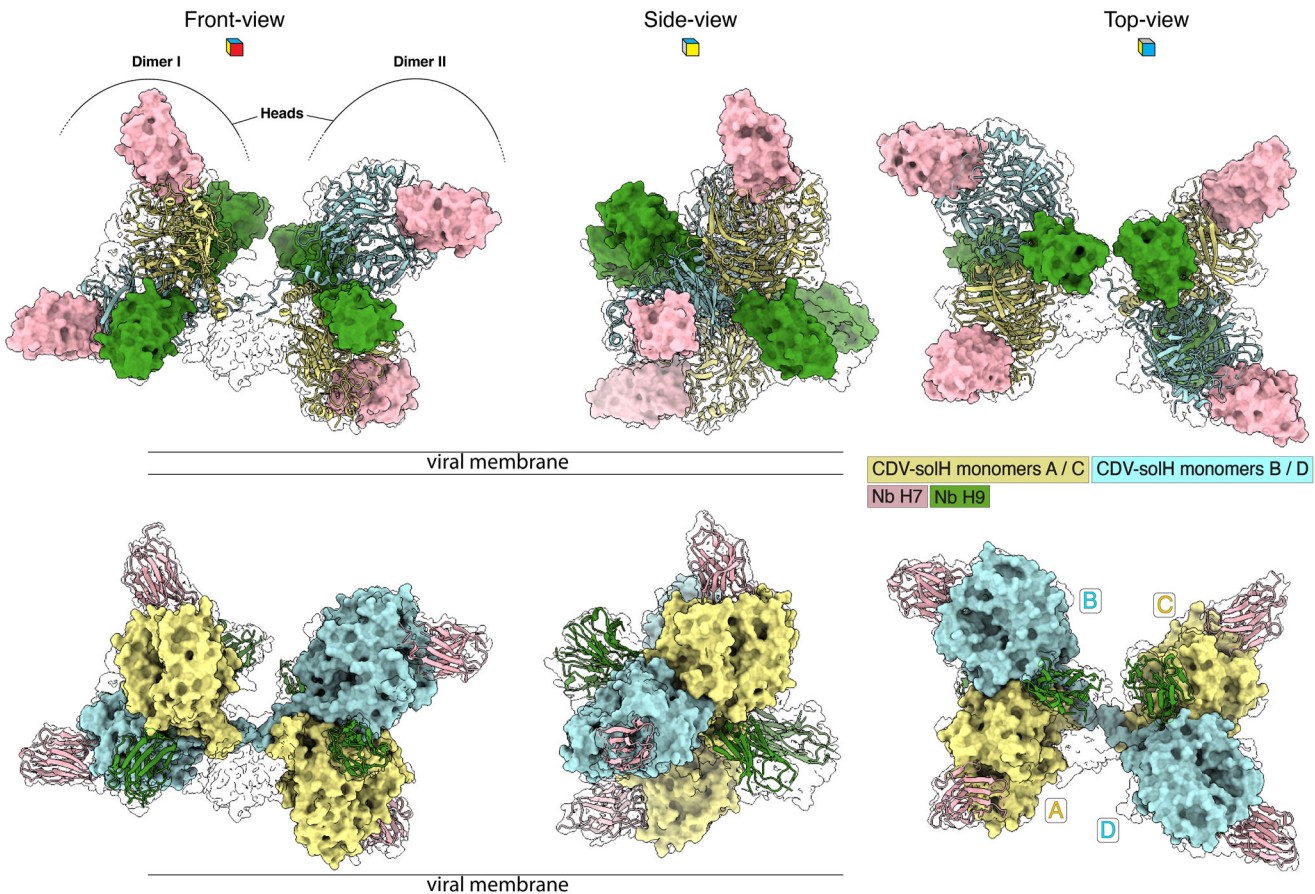

**Fig. 2 | Structure and supramolecular organization of the tetrameric CDV-solH bound with nanobodies H7 and H9.** The structure of tetrameric CDV-solH in complex with nanobodies (Nbs) H7 and H9 is displayed in different orientations relative to the putative viral membrane. The figure contains two panels (upper and lower), each showing different views to illustrate the architecture of the complex. In the upper panels, Nb-structures are represented as surfaces and CDV-solH head domains as ribbon; in the lower panels, the Nb-structures are displayed as ribbon and CDV-solH head domains as surfaces. The semi-transparent outline denotes the cryo-EM density map. In the upper front-view, the dimeric H-head domains are indicated. Individual proteins and protomers are color coded: Nb H7 (pink), Nb H9 (green), CDV-solH monomers A / C (pale yellow) and CDV-solH monomers B / D (light blue). Note that Nb H7 binds a single CDV-solH monomer, whereas Nb H9 engages the cleft at the interface of a CDV-solH dimeric head domain.

simultaneously to the CDV receptor-binding H protein. Moreover, while Nb H7 neutralized viral infections by competing with the SLAM receptor for the binding to CDV H, Nb H9 presumably blocked viral entry through a different, yet to be determined, molecular mechanism.

## Cryo-EM structure of the CDV H protein in complex with the neutralizing Nbs H7 and H9

To unveil the molecular basis of Nb H7 and Nb H9 neutralization, we performed single-particle cryo-EM analysis and 3D reconstruction of the purified solH-Nbs complex. A Coulomb potential density map (herein referred to as density) of the complex was first obtained at a global resolution of 4.3 Å. Importantly, local refinement of the dimers from the tetramer resulted in two maps, one at 3.1 Å for dimer I and the other at 3.4 Å for dimer II (Supplementary Figs. 2 and 3). The complex revealed eight Nbs bound to a single H-tetramer (Fig. 2 and Supplementary Fig. 4a, b). Since the neck and stalk domains were insufficiently resolved, an atomic model was built exclusively into the densities of the head domains and the eight neutralizing Nbs (Fig. 2 and Supplementary Fig. 4a, b). Compared to the previously published CDV-solH structure[12], the overall architecture of the head domain remained mostly unchanged in the presence or absence of the Nbs: four globular β-propeller head domains (each consisting of six β-sheets [blades]; Supplementary Fig. 5a) are organized in two non-contacting and relative to each other tilted dimeric head units (Fig. 2). The structure revealed two Nbs H7 and two Nbs H9 bound per dimeric

head unit (Supplementary Fig. 4c). Importantly, the model also unambiguously highlighted that the neutralizing Nbs H7 and H9 engaged two spatially distinct, non-overlapping epitopes, thus spotlighting two sites of vulnerability within the CDV H receptor-binding protein (Fig. 2 and Supplementary Fig. 4). Supplementary Fig. 6 illustrates two variations of amino acid numbering for both nanobodies, as well as an amino acid sequence alignment with indicated domains. We note that for this study (and corroborating our PDB entry), we selected the first option to describe the solH-Nbs interactions (Supplementary Fig. 6a).

## Nb H7 binds to an epitope overlapping with the putative receptor binding sites

Nb H7 bound into a cavity of a single H-head domain with a contact surface of ~1020 Å² (Fig. 3a). We defined the latter surface on the CDV H protein as "site I" neutralizing epitope. Nb H7 targets amino acids located in the β-propeller blades β–3, β–4 and β–5 of one head domain (Supplementary Fig. 5a, b). The β-sheets were labeled in accordance with those employed for the MeV H protein[65]. Overall, site I is composed of polar and hydrophobic interactions of the Nb H7 with the CDV H protein (Fig. 3a–c and Supplementary Data 1). Interestingly, Nb H7 harbors a long CDR1 domain that contains a significant number of residues critical for the interaction, while residues located in CDR2 and CDR3 are involved to a lesser extent. N79 is also part of the paratope but is located outside of the three CDRs (Fig. 3b, Supplementary Fig. 6

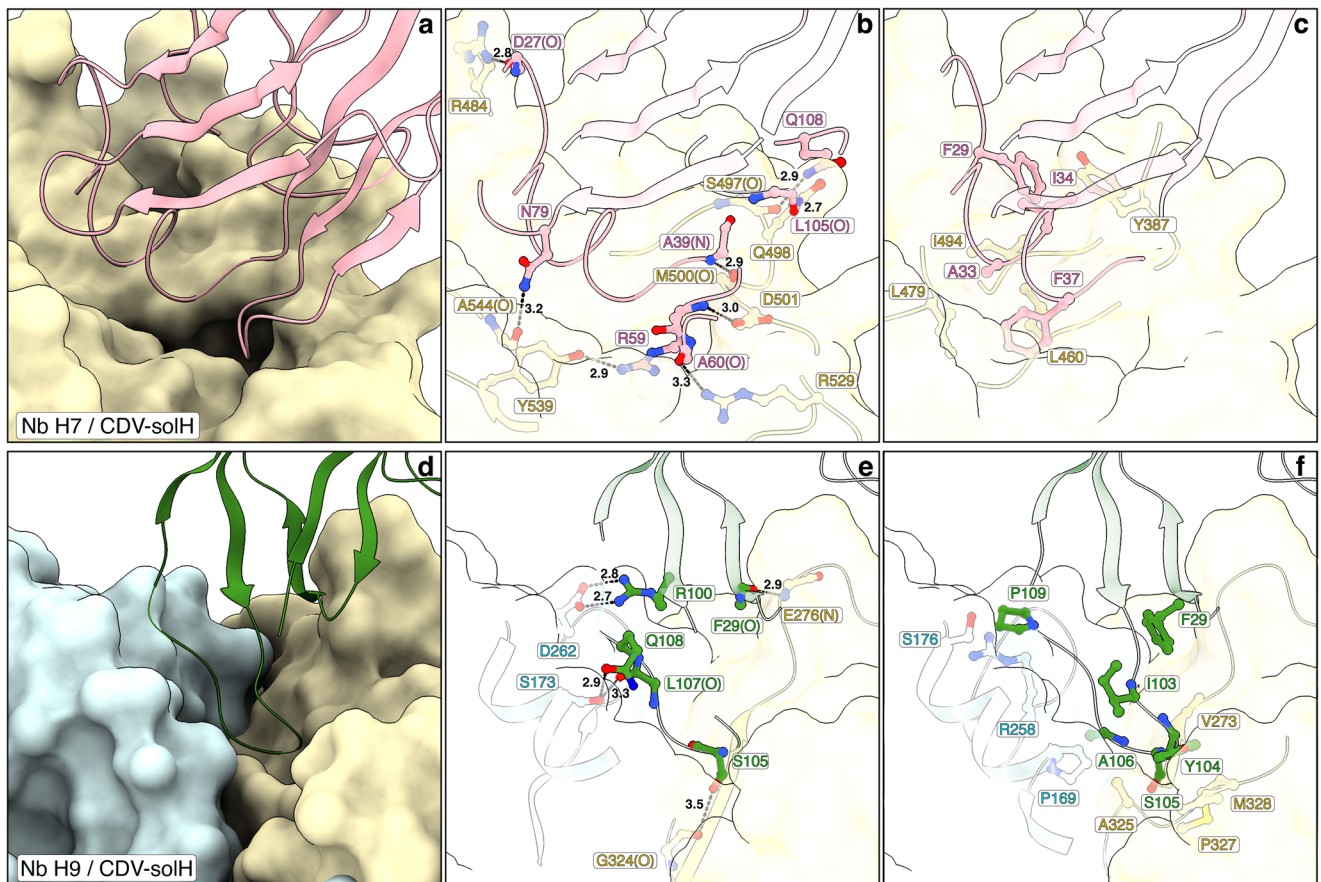

**Fig. 3 | The CDV-solH / Nb interfaces.** Interaction interfaces of CDV-solH with Nb H7 (site I; **a**–**c**) and Nb H9 (site II; **d**–**f**). Panels **a** and **d** show overviews; **b**, **e** highlight polar interactions and **c**, **f** hydrophobic ones. Color code: Nb H7 (pink), Nb H9 (green), CDV-solH-monomers A / C (pale yellow) and B /D (light blue). **b**, **c**, **e**, **f** Interacting amino acid residues within ≤ 3.5 Å (polar) and ≤ 4.0 Å (hydrophobic) are displayed as sticks and labeled (one letter code). When amino acid residues are interacting with their main-chain carbonyl oxygen or nitrogen atom, then they are additionally labeled with (O) or (N). Hydrogen bonds and salt bridges in panels (**b**, **e**) are highlighted as black dashed lines and distances are indicated in Å. Note that in panels (**b**, **c**, **e**, **f**) for reasons of clarity, some parts of CDV-solH, Nb H7 and Nb H9 are omitted.

and Supplementary Data 1). Furthermore, CDR1 encompasses four residues, i.e., F29, A33, I34 and F37, which determine hydrophobic interactions with residues located in a loop of blade β−4 of the H protein (Fig. 3c, Supplementary Fig. 5a, b and Supplementary Data 1). Importantly, "site I" overlaps with the putative canine SLAM and nectin-4 binding sites[12] (Supplementary Fig. 5b, d–f). The structure thus validated our functional data, suggesting that Nb H7 neutralized viral infection by competing with the CDV H/receptor interaction (Fig. 1e).

### Nb H9 binds at the dimeric head interface

In sharp contrast to Nb H7, Nb H9 bound deep into a cleft located at the interface between two protomers of the dimeric head unit (Figs. 2 and 3d). The quaternary epitope of Nb H9 spans a surface of 461 Å² on one protomer and 332 Å² on the other one. Consequently, we defined as "site II" the Nb H9 neutralizing epitope. Site II is composed of polar and hydrophobic interactions of the Nb H9 with the CDV H protein (Supplementary Data 1). These interactions encompass residues from the long CDR3 as well as residue F29 of CDR1 (Supplementary Fig. 6 and Supplementary Data 1). Notably, residues P169, S173, S176, R258, and D262 located in the head-to-neck connecting α-helix and blade β−1 of one protomer, together with residues V273, E276, G324, A325, P327 and M328 located in blades β−1 and β−2 of the other protomer define the quaternary epitope of Nb H9 (Fig. 3d–f and Supplementary Fig. 5a, c and Supplementary Data 1). Remarkably, the epitope is located far away from the one of Nb H7 and thus, as well as

from the putative SLAM and nectin-4 binding sites (Supplementary Fig. 5c, d, e, g). This also supports our experimental approach, which illustrated no detectable competition of Nb H9 with the H protein/ receptor interaction (Fig. 1e). Collectively, both functional and structural data provided evidence that Nb H9 employed a mode of neutralization that differs from blocking the RBS. The structure also highlighted that at the level of the H-tetramer, two Nb H9, each binding to distinct dimeric head units, are in relatively close spatial proximity (Fig. 2, top-view).

### Improved neutralization activity through the rational design of a biparatopic nanobody construct

Biophysical, functional and structural data suggested that fusing both Nbs together may translate into a significant enhancement in neutralization activity, i.e., binding to non-overlapping epitopes on CDV H and neutralizing through distinct mechanisms of inhibition. We thus engineered the H7-H9 biparatopic molecule, which consisted of Nb H7 fused to Nb H9 via a flexible (GGGGS)$_4$ linker (Supplementary Fig. 7a). The linker length is predicted to be long enough to enable the simultaneous binding of the H7 and H9 paratopes for the closer interaction possible within one dimeric head unit. Conversely, it is predicted too short for the second interaction within one dimeric head as well as for interactions between adjacent dimeric heads of the same H-tetramer (Supplementary Fig. 7b).

We initially expressed in bacteria, a His-tagged version of the biparatopic molecule, which was subsequently purified by metal

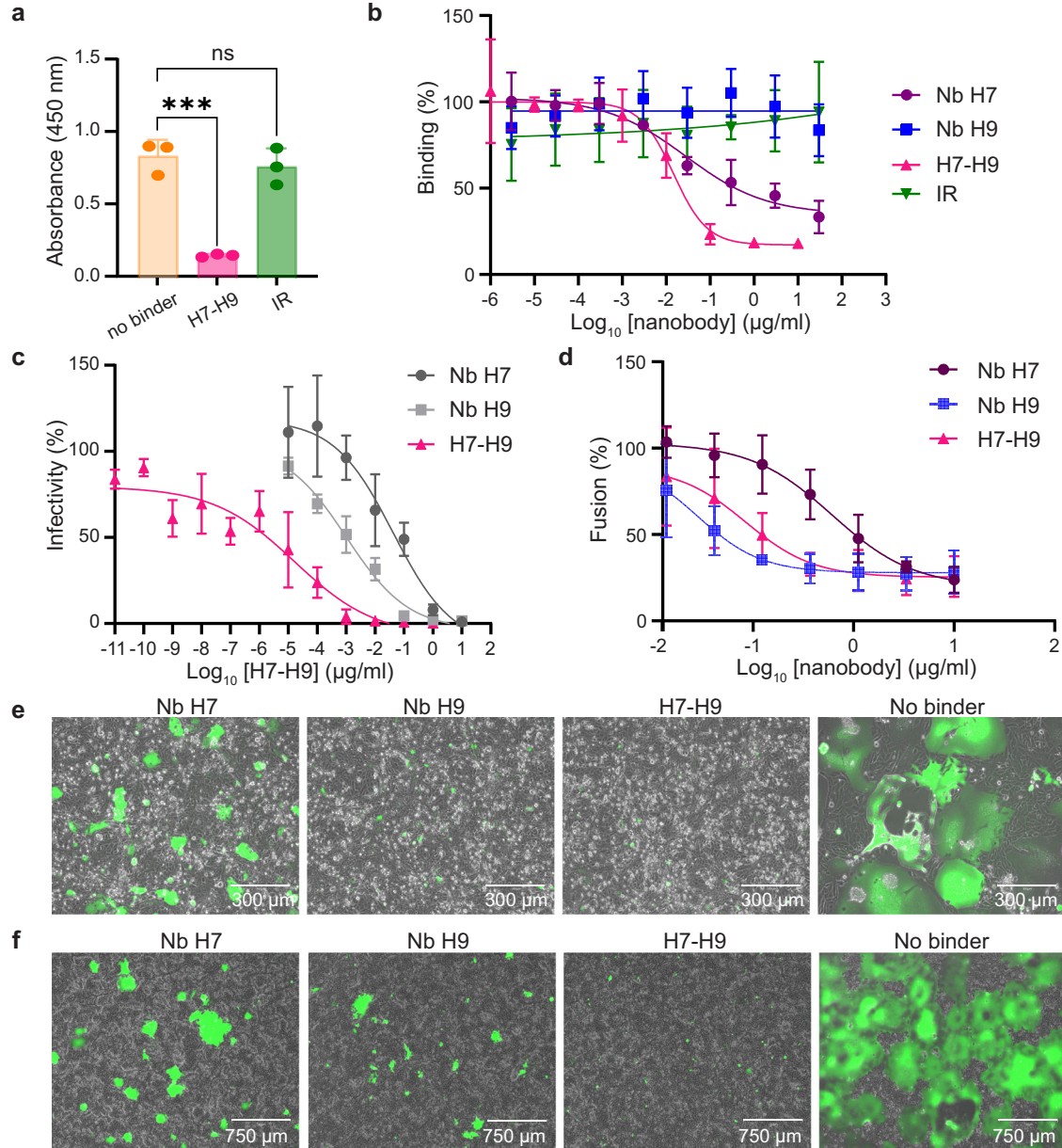

**Fig. 4 | Characterization of the H7-H9 biparatopic nanobody molecule.**
**a** Competition of the H7-H9 biparatopic nanobody (Nb) construct and SLAM to CDV H binding, investigated by ELISA. The recombinant solH protein was immobilized on a plate and followed by addition of H7-H9 or an irrelevant (IR) Nb, respectively (indicated on the x-axis). Subsequently, plates were exposed to a soluble Fc-carrying version of SLAM. Data are mean ± SD calculated from $n = 3$ independent experiments performed in 3 technical replicates (with binders) and in 2 technical replicates (without binders). Statistical significance was determined using a one-way ANOVA followed by Dunnett's multiple-comparison test, using GraphPad Prism v.9 (***$P \leq 0.001$; ns: non-significant). **b** Similar experimental setup as described in (**a**) but with the addition of serial dilution of the concentrations of the Nb H7, H9, H7-H9, or an irrelevant (IR) Nb. Data are mean ± SD calculated from $n = 3$ independent experiments performed in technical triplicates. **c** Neutralization assays against CDV-A75/17$^{neon/nlucP}$. The panel depicts the relative viral infectivity at increasing concentrations of the biparatopic Nb H7-H9 (pink line), and, as

comparison, the relative viral infectivity at increasing concentrations of the monovalent nanobodies (light and dark grey lines). Data are mean ± SD calculated from $n = 3$ independent experiments performed in technical triplicates. **d** Quantitative cell-to-cell fusion inhibition assays. The indicated Nbs were added four hours post transfection of Vero cells with plasmids encoding the CDV H and F proteins, and LgBiT-GFP. In parallel, Vero-cSLAM cells were transfected with plasmid expressing HiBiT-RFP. Eighteen hours post cell population-mixing, cell-cell fusion was indirectly measured by recording luciferase activity. Data are mean ± SD calculated from $n = 3$ independent experiments performed in technical triplicates. **e, f** Qualitative cell-to-cell fusion inhibition assays. The indicated Nbs were added four hours **e** post transfection of Vero-cSLAM cells with plasmids expressing the CDV H and F proteins, and GFP proteins, or **f** post infection of Vero-cSLAM cells with CDV A75/17$^{neon/nlucP}$. Cell-to-cell fusion was recorded by taking images of the fluorescence emission or by recording the luciferase activity, respectively. Curves (**b–d**) were plotted with GraphPad Prism v.9.

affinity chromatography and size-exclusion chromatography. We noted that H7-H9 efficiently bound to the membrane-anchored form of CDV H (Supplementary Fig. 1a) and exhibited an approximately 2 to 3-fold increase in binding affinity to solH, relative to both monovalent Nbs, as revealed by SPR analyzes ($K_D$ of 2.3 nM and 5.4 for Nb H7 and H9, respectively, and 1.9 nM for H7-H9) (Supplementary Fig. 1e and

Table 1). As expected from data collected with the Nb H7, the biparatopic construct also significantly competed with the SLAM (Fig. 4a). To compare the competition efficiencies of H7 and H7-H9 more accurately, we repeated the SLAM competition assay but with serial dilutions of the Nbs. We noted that the biparatopic construct performed substantially better than the monovalent Nb H7 (IC$_{50}$ values of

about 0.015 and 0.028 μg/ml [0.49 and 1.8 nM] for H7-H9 and H7, respectively) (Fig. 4b). Furthermore, and most remarkably, H7-H9 exhibited sub-picomolar $IC_{50}$ values in neutralization activities (about 0.6 pM) (Fig. 4c and Table 1), which represented an approximately ≥100-fold improvement over Nb H9, and about >4000-fold over Nb H7.

Membrane fusion plays a key role not only in virus-cell entry but also in lateral cell-to-cell spread, leading to the formation of multinucleated cells known as syncytia. Syncytia formation may also facilitate viral propagation within the host[18]. To assess whether the individual single-domain antibodies as well as the highly potent biparatopic construct could also inhibit cell-to-cell fusion, we performed membrane fusion assays. Strikingly, compared to untreated cells, treatments of H/F/GFP-transfected as well as A75/17[neon/nlucP]-infected Vero-cSLAM cells with 5 μg/ml of Nb H7, H9 or the biparatopic H7-H9 molecule (added 4 h post transfection or infection) resulted in a strong inhibition of cell-to-cell fusion, as revealed by quantitative (Fig. 4d) and qualitative (Fig. 4e, f) fusion assays. We also assessed the binding efficiency of the biparatopic H7-H9 Nb construct to the MeV H protein in ELISA. Although binding activity was recorded at very high concentration, the efficiency was reduced by at least three orders of magnitude compared with its binding to CDV H protein (Supplementary Fig. 1d).

Taken together, these data provided evidence that the structure-guided H7-H9 biparatopic construct led to the generation of a robust neutralizing molecule; a high potency that may result from a combination of increased avidity to CDV H (biparatopic construct) and the integration of multiple mechanisms of inhibition in a single format.

## Combining both nanobodies restores inhibition against CDV H proteins carrying resistance mutations

To determine the inhibitory effect of both Nbs against H proteins carrying mutations found in selected circulating CDV strains (as well as the vaccine strain Ondersteeport large-plaque forming variant (OL-CDV)), we substituted residues in the A75/17-CDV H protein (H-wt) that differed from the selected strains and present in the epitope targeted by Nb H7 (L460V, R484W, M500I, A544S and A544T) and Nb H9 (protomer 1: D262N; protomer 2: V273I, E276V, G324W and A325G) (Supplementary Data 2 and 3). We employed a quantitative membrane fusion-inhibition assay, which allowed for an accurate comparison of both Nbs activity against the series of H mutants. Transfected cells were treated with 10, 0.5 and 0.025 μg/ml of the various corresponding nanobodies or left untreated as control. The results revealed that, compared to H-wt, Nb H9 exhibited very similar efficacy against CDV H proteins harboring mutations affecting the epitope of Nb H7. Similar data were recorded in the reversed situation (Nb H7 on H proteins harboring mutations in the H9-targeting epitope). Conversely, the substitution G324W dramatically impaired Nb H9 efficacy, whereas the mutation M500I, exclusively found in the vaccine OL-CDV strain, significantly impacted Nb H7 potency. The double mutant H-G234W/A325G (found in the "South America 2" CDV lineage) exhibited the strongest escape profile towards Nb H9 (Supplementary Data 7). We next investigated the membrane fusion-inhibition potency of the biparatopic construct (H7-H9) with H proteins carrying the two mutations exhibiting the strongest resistance profiles (M500I and G324W). Membrane fusion-inhibition was indeed recorded in case of the single H-M500I and H-G324W variants. Although membrane fusion activity triggered by the double H-G324W/A325G mutant was strongly inhibited at the highest concentration of the biparatopic H7-H9 construct (10 μg/ml), partial activity was nevertheless recorded at the concentration of 0.5 μg/ml (Supplementary Data 7).

Overall, these data indicated that circulating CDV strains carrying the H mutations G324W may be less sensitive to Nb H9. Conversely, Nb H7 may exhibit *pan*-CDV activity, except for viral variants carrying the M500I mutation (e.g., the vaccine OL-CDV strain). Importantly, our

data additionally revealed that H proteins carrying resistance mutations against the monovalent nanobodies H7 or H9 were blocked by the biparatopic H7-H9 molecule. However, we cannot exclude that naturally occurring combinatorial mutations in circulating strains may compromise the broad neutralizing capability of the biparatopic H7-H9 construct.

## A tetravalent biparatopic antibody (H7-H9-fFc) protects ferrets against lethal CDV infection

Having established the robust neutralizing activity of the biparatopic H7-H9 molecule against the wild-type A75/17-CDV strain in vitro, we next evaluated its efficacy in vivo. We selected the ferret model of CDV infection, because the CDV strain 5804P is well-known to rapidly infect ferrets and cause fatal disease[66]. Because Nbs are known to be swiftly cleared from the bloodstream, we grafted the Fc domain of the ferret IgG molecule (fFc) to the biparatopic H7-H9 molecule (Supplementary Fig. 7c); a strategy demonstrated to significantly improve the stability of Nbs in the bloodstream as well as to potentially recruit Fc-dependent effector functions[63]. We additionally introduced a single point mutation in the fFc domain (S252Y), reported to further increase its stability[67]. The Fc fusion was produced in mammalian cells and followed by protein A-based purification. Proper Ab covalent dimerization was next confirmed by SDS-PAGE analyzes ran under reducing and nonreducing conditions (Supplementary Fig. 7d). The resultant tetravalent, biparatopic antibody (hereafter referred to as H7-H9-fFc) demonstrated very potent neutralizing activities against A75/17[neon/nlucP] in vitro, typically exhibiting $IC_{50}$ values of about 0.06 pM (Supplementary Fig. 7e and Table 1). To demonstrate that our rationally designed multidomain antibody effectively cross-neutralized the genetically closely related CDV strain 5804P in vitro, the effect of H7-H9-fFc was determined employing a standard progeny virus-inhibition assay. For this purpose, we infected Vero-cSLAM cells with wild-type A75/17-CDV, 5804P-CDV and as well as OL-CDV (a vaccine CDV strain also not expressing the Nanoluciferase reporter protein) in the presence of increasing concentrations of the antibody. One (A75/17 and 5804P) or two (OL) day(s) post-infection, the cells were subjected to three freeze-thaw cycles and the harvested viruses were titrated in Vero-cSLAM cells. Our findings revealed that H7-H9-fFc inhibited virus production of A75/17-CDV and 5804P-CDV in a very similar manner, even though the H protein of 5804P-CDV carries the D262N mutation (Supplementary Fig. 7f). Interestingly, although virus production of OL-CDV was also inhibited by H7-H9-fFc, the impact was much less pronounced (Supplementary Fig. 7f), as also inferred from our mutational analyzes (Supplementary Data 7).

For in vivo studies, four treatment groups of three ferrets each were given the biparatopic H7-H9-fFc construct according to the plan shown in Fig. 5a, or left untreated. Considering the high efficacy recorded in the neutralization assays, we selected a very low dose of H7-H9-fFc for all animal experiments (1 mg/kg), which is in a range similar to other studies[68,69]. As expected, infection with the wild type 5804P expressing GFP resulted in rapid onset of rash (Fig. 5b) and fever, often in excess of >40 °C (Fig. 5c and Supplementary Fig. 8a). All animals had detectable viremia starting on day 3, and then developed high viremia peaks 7 days after infection (Fig. 5d). When clinical endpoints were reached (from day 10), untreated animals were euthanized (Fig. 5e).

In sharp contrast, all animals treated prophylactically with H7-H9-fFc (4x and 3x treatment groups) survived the swift lethal disease induced by the highly pathogenic strain. While viremia was not detected on day 3, viral titers increased moderately between days 7 and 10 (Fig. 5d). All animals in the 3× treatment group had cleared the virus by day 21, and 2 out of 3 animals in the 4× treatment group cleared the virus after day 17 (Fig. 5d). Although no signs of rash or substantial increase in body temperature was monitored, one animal of this group did not successfully clear the virus (Fig. 5b–d). Consequently, between days 28–30 post infection, the animal developed

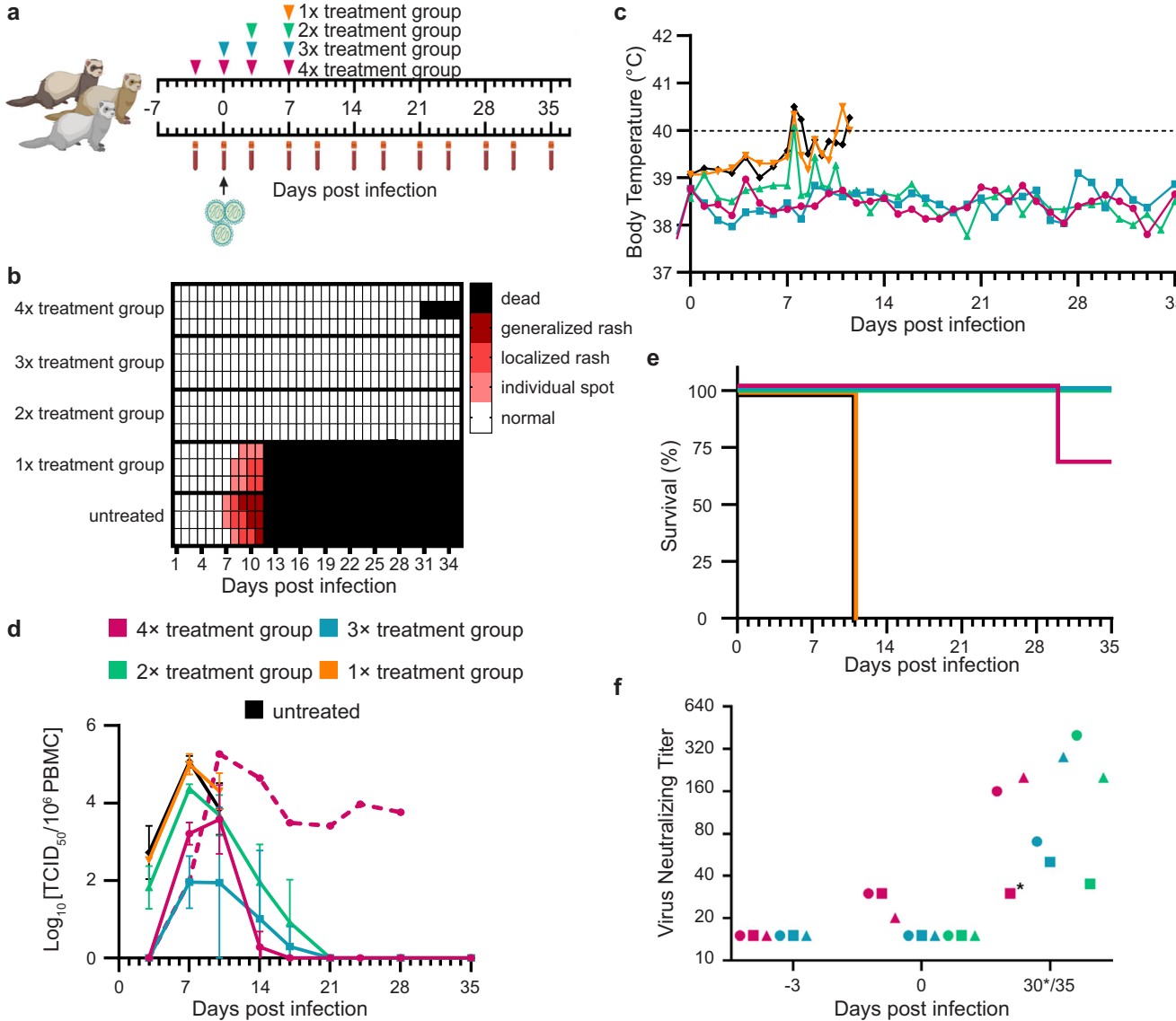

**Fig. 5 | A tetravalent, biparatopic antibody (H7-H9-fFc) protects ferrets against lethal CDV infection. a** Antibody treatment timeline. Groups of three ferrets (*n* = 3 biological replicates) were treated according to the plan shown on top. Blood sampling times and the day of infection are indicated under the timeline. Created in BioRender. Sawatsky, B. (2026) https://BioRender.com/jgzifhx. **b** Development of rash. The rash severity of each animal is represented by one row. The color of each square indicates the severity (white, no rash; light red, individual spots; red, localized patches of rash; dark red, generalized rash; black, death of an animal). **c** Body temperature of infected animals. Mean daily rectal temperatures are shown for each treatment group. The dashed line at 40 °C indicates the threshold for high fever. **d** Viral titers in peripheral blood mononuclear cells (PBMCs) of infected animals. Titers are shown as the $\log_{10}$ tissue culture infectious dose (TCID)$_{50}$ per $10^6$ PBMCs. Mean viral titers are shown for each treatment group and error bars indicate standard deviations. One animal (purple dashed line) did not clear the virus, developed neurological signs of disease, and was euthanized on day 30. **e** Survival of ferrets. The death of an individual animal is shown as a step down on the curve. **f** CDV neutralizing responses in plasma. Antibody titers against CDV 5804P are shown as the reciprocal of the highest dilution at which cytopathic effect (CPE) was observed. Each animal is represented by a symbol whose corresponding treatment group color as shown in (**a**). The animal indicated with an asterisk (*) developed neurological signs of disease (ataxia, lack of hind limb coordination, and difficulty walking) and was euthanized on day 30.

ataxia, lack of hind limb coordination and difficulty walking, which are signs of neurological complications. The animal was euthanized on day 30 (Fig. 5e) and evidence of viral infection of the brain was validated upon post-mortem examination. Several animals developed transient mild to moderate fever (between 39 °C and 40 °C), which then resolved quickly (Fig. 5c and Supplementary Fig. 8a). The body temperatures of the other animals remained within normal ranges. Thus, administration of prophylactic doses of H7-H9-fFc were highly effective at preventing the development of lethal disease after infection with the wild type 5804P-CDV strain.

There was a marked difference in outcomes between the two groups of ferrets treated after infection (2× and 1× treatment groups).

Remarkably, although animals in the 2× treatment group (treated on days 3 and 7) had substantial viremia on days 3 and 7 after infection, it resolved afterward, and all animals cleared the virus by day 21 (Fig. 5d). They experienced mild to moderate fever of 39–40 °C, but otherwise showed no outward signs of clinical disease (Fig. 5b, c and Supplementary Fig. 8a). Conversely, animals in the 1× treatment group (treated on day 7) experienced identical course of viremia (Fig. 5d), but developed clinical signs, such as generalized rash (Fig. 5b), conjunctivitis, weight loss (Supplementary Fig. 8b, c), and sustained fever (>40 °C) for several days (Fig. 5c and Supplementary Fig. 8a). Clinical endpoints were ultimately reached after day 10 and the animals were euthanized (Fig. 5e). Collectively, although a single treatment with H7-

H9-fFc on day 7 post infection had no significant impact on the lethal disease outcome, two therapeutic injections of H7-H9-fFc (days 3 and 7 post infection) at a concentration of 1 mg/kg fully protected and prevented the development of clinical signs induced by the wild type 5804P-CDV strain.

Prolonged suppression of virus replication can result in delayed or impaired immune responses to infection[38]. We tested the sera of animals to determine whether H7-H9-fFc treatment had similar effects on the development of neutralizing antibody responses. All animals that survived the infection developed neutralizing antibodies that would be protective against subsequent lethal CDV infection (Fig. 5f)[70]. The neutralizing antibody titer from the ferret in the 4× treatment group that succumbed to neurological disease (marked with an asterisk in Fig. 5f) would also likely protect against challenge in this respect[71].

### The H7-H9-fFc antibody neutralizes the CDV brain isolate

Eight out of nine ferrets that received at least two H7-H9-fFc injections efficiently cleared the virus with no severe clinical signs, and no evidence of resistance was observed. To explore whether the ferret that exhibited brain infection correlated with the emergence of resistance mutations, the amino acid sequence of the receptor-binding H protein was determined from RNA extracted from peripheral blood mononuclear cells (PBMCs) at several time points post-infection as well as from brain cells. We identified a serine to tyrosine substitution at position 546. The mutation emerged post-treatment: between days 7 and 14 post infection, and was found in both PBMCs and brain cells. The mutation is located in close proximity to the SLAM binding site and, therefore, also adjacent to the Nb H7 epitope. Tyrosine 546 may induce clashes with neighboring residues in the CDV H protein that are in contact with Nb H7 or may mediate long-range effects impacting the potency of Nb H7. We thus investigated the ability of Nb H7, H9 and H7-H9-fFc to impair progeny virus production of the brain isolate. While Nb H7 exhibited reduced efficacy against the brain isolate as compared to the wild type 5804P CDV strain (Supplementary Fig. 9a), the monovalent Nb H9 (Supplementary Fig. 9b) as well as the tetravalent and biparatopic antibody (Supplementary Fig. 9c) neutralized both viruses with equal efficacies. Similar results were obtained using our quantitative membrane fusion-inhibition assay (Supplementary Data 7).

In summary, our data revealed that the emergence of the S546Y substitution in the H protein, which appeared post-treatment, did not negatively affect the efficacy of our engineered Ab in the context of neutralization assays.

## Discussion

The treatment landscape for infectious diseases is evolving significantly with the rise of monoclonal antibodies. However, a key limitation of mAbs is their high production cost, especially when multiple antibodies are required to achieve an acceptable clinical profile, which limits their use mainly to high-income countries. Even in these countries, healthcare systems start facing increasing challenges for covering mAb and other expensive therapies for insured patients due to rising costs. In contrast, llama-derived nanobodies offer a promising alternative as antiviral drugs and beyond due to their small size, favorable physicochemical properties and ease of formatting. Notably, a single Nb-based multiparatopic molecule can target multiple vulnerable sites on viral glycoproteins simultaneously, thereby not only increasing the apparent affinity for its target and lowering the risk of the emergence of resistant viral variants, but meanwhile offering the advantage of significantly reduced production costs.

To establish a roadmap for the development of new treatment options against morbilliviral infections (e.g., MeV or CDV), we identified two Nbs (H7 and H9) that efficiently bound to the CDV receptor binding (H) protein and potently neutralized the infection. Biophysical and functional analyzes indicated that Nb H7 competed with receptor

interaction, whereas Nb H9 blocked viral entry through a different mechanism. Cryo-EM analyzes revealed their mode of interaction at high resolution and validated that both Nbs bound to CDV H at two non-overlapping epitopes, thereby highlighting two vulnerable sites of neutralization. Site I, targeted by Nb H7, largely overlaps with the putative RBSs of SLAM and nectin-4[12], suggesting its competition with these receptors, as confirmed by competition ELISA using soluble SLAM. On the other hand, Nb H9 bound to site II, which encompasses amino acids located in two protomers at the dimeric head interface, thereby defining a complex quaternary epitope. As already inferred from the spatially remote locations of site I/RBS and site II, Nb H9 did not compete with SLAM interaction. Although the mechanism of neutralization mediated by H9 remains currently undefined, we speculate that Nb H9 may interfere with a conformational change in H that is required to activate F trimeric complexes upon H head(s)-mediated receptor contacts. Alternatively, Nb H9 may interfere with F interaction.

Remarkably, combining Nbs H7 and H9 in a single format (with or without Fc) resulted in molecules with significantly enhanced neutralization profiles, and surpassing the activity of both monovalent Nbs added together. Beyond the increased avidity of interaction, we hypothesize that a supplementary mechanism of inhibition might have contributed to the overall high potency of the engineered biparatopic constructs. Indeed, the length of the flexible linker that fuses both Nbs together was rationally designed based on our cryo-EM structure to potentially enable intra but not interdimeric head unit binding. However, simultaneous interaction of the biparatopic constructs to both dimeric head units may have nonetheless occurred by inducing untimely rearrangements of the H heads. This neutralizing mechanism may be remarkably effective since a single biparatopic molecule per CDV H tetramer may be sufficient to trigger full neutralization.

The tetravalent, biparatopic H7-H9-fFc antibody demonstrated prophylactic and therapeutic protection of ferrets infected with a lethal CDV strain upon administration of only 1 mg/kg. Animals treated with H7-H9-fFc on days 3 and 7 after infection not only survived but did so without developing overt signs of clinical CDV disease. Morbillivirus diseases are best thought of as a race between the virus and the host immune system, where early viral replication overwhelms the immune system allowing systemic spread to other organ systems[72]. Therapies that hinder virus replication to a sufficient extent as early as possible can allow the host immune response to "catch up"[38]. Of note, the animal in the 4× treatment group, which developed neurological disease, had higher viremia than the other animals, but did not exhibit any outwards clinical signs in the early phase of the infection. The ferret was ultimately not able to successfully clear the virus, which persisted in the blood at a relatively elevated level after the last treatment on day 7. The duration of viremia and disease are known to be critical factors for neuroinvasion and the development of neurological disease[73]. The 5804P strain of CDV quickly invades lymphoid tissues and rapidly causes generalized immunosuppression and severe disease. Therefore, although it does not normally cause neurological disorders, sustained viremia over several weeks of infection might have allowed the virus to eventually infiltrate the brain, at which point incurable brain disease develops and peripheral immune responses are ineffective. Further supporting this hypothesis, a recombinant neuroinvasive CDV strain (A75/17) harboring the 5804P H protein was reported to maintain efficient brain infection[73]. We nonetheless identified a point mutation in the H protein (S546Y), which emerged post-treatment (between days 7 and 14 post infection) in both infected brain cells and PBMCs. While progeny virus production by the brain isolate showed decreased sensitivity to Nb H7, Nb H9 remained fully effective. Importantly, H7-H9-fFc inhibited virus production of both the original virus and the brain isolate with equal efficacy. Hence, it is tempting to speculate that administering an additional dose of H7-H9-fFc after day 7 (which was fixed in our experimental protocol) could have further

decreased viral titers in the blood and mitigated central nervous system invasion. Notably, recent reports have shown that analogous Nb-based multivalent anti-spike constructs maintained efficacy against several SARS-CoV-2 variants, despite these variants being resistant to the Nb in their monovalent format[74]. This suggests that the collective binding avidity of a multidomain Ab may contribute to the sustained efficacy, compensating for potential limitations observed with monovalent formats.

In low-income countries, vaccination of dogs against CDV remains suboptimal and distemper remains a significant animal health issue. Additionally, in captive animals, CDV infections are a recurring problem and the only solutions thus far have been emergency vaccination or supportive therapy. In some highly susceptible species, the live-attenuated vaccine itself can be problematic. Pandas and fennec foxes, for instance, are prone to developing disease after vaccination, which can be fatal[75,76]. Our multiparatopic antibody could conceivably be an alternative or even employed in combination with those standard therapies. Importantly, we have shown here that treated animals develop strong neutralizing antibody titers after virus challenge, so this approach would have the added benefit of inducing long-term protective immunity against CDV in vulnerable animals.

The amino acids contributing to the site I and site II neutralization epitopes are well conserved among various CDV strains (Supplementary Data 2, 3 and 4). When signature substitutions of circulating CDV strains were introduced into the H protein of the A75/17 CDV strain, resistance mutations were identified against both monovalent Nbs. Nevertheless, combining both Nbs in a single format successfully inhibited those H variants (Supplementary Data 7). Although this must be formally confirmed in preclinical studies, this infers broad efficacy of our engineered, multidomain Ab at least against contemporary circulating CDV strains. In contrast, the receptor binding H protein of MeV shows only 35% amino acid sequence identity as compared to the one of CDV and, indeed, multiple residues vary within both neutralizing epitopes (Supplementary Data 2, 3 and 5), which explains the lack of reactivity of both monovalent Nbs towards the MeV H protein (Supplementary Fig. 1d). Although the biparatopic H7-H9 construct exhibited improved binding activity to MeV H, we are convinced that high affinity binders to MeV H must be discovered to develop viable Nb-based antiviral drugs. Alternatively, the cryo-EM structure of the CDV H glycoprotein in complex with both neutralizing Nbs provides a template not only to improve the binding affinity of both Nbs to CDV H, but may be employed to potentially extent the breadth of the Nbs towards other morbilliviruses (e.g., MeV).

In summary, our findings demonstrate that an engineered antibody, designed to expose two nanobodies that simultaneously target distinct viral antigenic sites and neutralize through different molecular mechanisms, effectively protected ferrets infected with a lethal CDV strain. Overall, we present a cost-effective antiviral strategy that may be applied to combat other pathogens of significant medical importance.

## Methods

### Ethics statement
Ferret experiments were conducted at the Paul-Ehrlich-Institute in accordance with all applicable German and European regulations under protocol number V54-19c20/15-F107/1054 issued by the Regierungspräsidium Darmstadt.

### Cells, viruses and transient transfections
HEK-293T/17 (ATCC CRL-11268), Vero (ATCC CCL1-81), and the derivative cells stably expressing the canine SLAM receptor (Vero-cSLAM, kindly provided by Yusuke Yanagi, Kyushu University, Japan) were grown in Dulbecco's modified Eagle's medium (DMEM) (Gibco) at 37 °C in the presence of 5% $CO_2$. The DMEM was supplemented with 10% fetal calf serum (FCS) (PAN Biotech) and 1% penicillin-streptomycin. The recombinant CDV viruses used in this study were

amplified in Vero-cSLAM cells. These viruses included CDV wild type strain A75/17, containing dual reporter proteins (mNeonGreen (neon) and Nanoluciferase (nlucP)-expressing cassettes inserted between the P and M genes[77] (A75/17[neon/nlucP]), CDV wild type strain 5804P, containing a green fluorescent protein (GFP)-expressing cassette inserted between the H and L genes (5804P-GFP[78]), CDV wild type strain A75/17, and CDV vaccine strain Onderstepoort large plaques, containing a GFP-expressing cassette inserted between the H and L genes (OL-GFP). Transient transfections of the plasmids expressing A75/17-CDV H protein, F protein, and GFP were carried out by employing the TransIT-LT1 Transfection Reagent following the manufacturer's instructions (Mirus).

### Expression and purification of CDV H protein ectodomain
The CDV H protein ectodomain (solH) was designed as described previously[12]. Briefly, the solH construct contains, from the N- to the C-terminus, the following elements: the mouse Ig Kappa signal peptide, a 8xHis/Twin-Strep-tag[79] (TST), a tetramerization GCN4 motif[80], and the CDV H residues 60 to 607 (strain A75/17). To ensure efficient protein production, solH was produced at the protein-expression core facility of the École Polytechnique Fédérale de Lausanne (EPFL, Switzerland; Protein Production and Structure Core Facility). Approximately $2 \times 10^9$ expiCHO cells grown in suspension were transfected with 3 mg of the expression plasmid. After a 7-day expression period at 37 °C, 1 L of supernatant was harvested. Subsequently, the protein was purified from the supernatant by using a 5 mL StrepTtrapXT column (Cytiva), utilizing 100 mM Tris-HCl pH 8.0, 150 mM NaCl, and 1 mM ethylenediaminetetraacetic acid (EDTA) as the buffer. Protein elution was achieved in the same buffer, supplemented with 50 mM biotin (IBA-Lifesciences). A similar solH construct derived from MeV (IC-B strain) was also engineered (i.e., TST-GCN4-MeV H-ectodomain). The ectodomain consists of amino acids 60 to 617. Expression and purification steps of MeV solH were performed according to the protocol mentioned above for CDV solH.

### Generation and identification of Nbs targeting the CDV H glycoprotein
A Nb library was constructed in the *Escherichia coli* TG1 strain after immunisation of a llama (*Lama glama*) with the recombinant soluble using 4 injections of 280 µg of tetrameric-stabilized, CDV H protein ectodomain (solH)[81] mixed with incomplete freund's adjuvant over a period of 6 weeks. Immunizations were outsourced in 2017 by employing the services of Ardèche Lama (a llama breeding farm) in Saint-Remèze (France). Immunizations were performed by a professional veterinarian and were executed in strict accordance with good animal practices, following the EU animal welfare legislation law. Briefly, VHH genes were amplified by a two RT-PCR using 50 µg of total RNA extracted from PBMCs as template, digested and cloned into the phagemid pHEN1 to generate a library of >10[8] transformants. This library was rescued using helper phage KM13. Two different strategies were used during this subsequent selection by phage display, SolH was either immobilised (2.5 µg/ml) on 96-well plates (MaxiSorp, NUNC) or coated (42.5 µg/ml) on epoxy bead M-450 (Dynabeads M-450 Epoxy, Invitrogen). This was followed by two rounds of selections. Then, 372 individual TG1 colonies were randomly picked for each strategy and grown overnight at 37 °C in 2YTAG (2YT complemented with 100 µg/ml ampicillin and 2% (w/v) glucose). Overnight cultures were used to inoculate fresh 2YTA medium. After growing for 2 h at 37 °C, the production of Nbs was induced by the addition of 100 µM isopropyl β-D-1-thiogalactopyranoside (IPTG) (Merck) and overnight growth at 30 °C. Supernatants were harvested and used for a first screening against solH immobilized (2.5 µg/ml) on MaxiSorp 96-well plates. The binding of Nbs was detected with an anti-His antibody (Novagen, 1:1000 dilution) and horseradish peroxidase (HRP)-conjugated goat anti-mouse IgG (GAM-HRP, 1:5000 dilution (Jackson ImmunoResearch). A second

screening was realized on positive clones obtained from the first screening to confirm the binding on membrane-anchored CDV H protein. Binding was investigated either on CDV H stably expressing HEK 293T/17 cells or regular HEK 293T/17 cells. Binding of Nbs was detected using an anti-His antibody (Novagen, 1:1000 dilution) and Alexa 647-conjugated goat anti-mouse IgG (Alexa 647-GAM, 1:500 dilution) (Invitrogen). Genes of positive clones were sequenced.

### Expression and purification of Nbs

Anti-CDV H Nbs (cloned in the pHEN vector[82]), which harbor the c-Myc and 6xHis dual tags at their C-terminus, were transformed into *E. coli* BL21 (DE3) cells (Invitrogen) and cultured at 37 °C (180 rpm, Minitron INFORS HT; overnight) in 5 ml 2xYT medium (Sigma-Aldrich) containing 100 μg/ml ampicillin. Upon determination of the optical density ($OD_{600}$), the culture was diluted to an $OD_{600}$ of 0.1 in 250 ml fresh 2xYT medium, containing 100 μg/ml ampicillin, then incubated at 37 °C (180 rpm, Minitron INFORS HT) until an $OD_{600}$ of about 0.6 was reached. Protein expression was induced by adding IPTG to a final concentration of 100 μM. The culture was further incubated at 25 °C (180 rpm; overnight). The next day, the cells were harvested by centrifugation (10 min; 5000 × g) and the Nbs extracted from the periplasm. Briefly, the cell pellet was resuspended in lysis buffer (3 ml/g cells), containing 200 mM Tris-HCl pH 8.0, 500 mM sucrose, 1 mM EDTA and protease inhibitors (cOmplete Protease Inhibitor Cocktail, Roche), followed by incubation on ice for 30 min. Next, 4× volume of the previous lysis buffer in PBS, supplemented with 1 mM MgCl₂, was added and incubated on ice for another 30 min. After centrifugation for 15 min at 10,000 × g, the supernatant was collected and filtered through a 0.22 μm filter. Subsequently, Nbs were purified by using 1 ml HiTrapHP column (Cytiva), using 20 mM Tris-HCl pH 8.0, 500 mM NaCl, 20 mM imidazole. Nbs were eluted in the same buffer supplemented with 500 mM imidazole. Imidazole was then removed, and Nbs were rebuffered into PBS using PD-10 Desalting Columns (Cytiva) following the manufacturer's instructions.

### Design, expression and purification of engineered Nb constructs

The biparatopic Nb construct (H7-H9) was designed in silico and synthesized (Biocat). Briefly, the molecule consists, from the N- to C-terminus, of the following gene sequences: Nb H7, a flexible glycine/serine linker (GGGGS)₄, Nb H9 and a 6xHis-tag. H7-H9 was subsequently cloned into the pSBinit bacterial expression vector (Addgene). Production and purification steps were performed as indicated for the monomeric Nb binders, with minor modifications: chloramphenicol (25 μg/ml) was used and induction of protein expression was performed with 0.02% (w/v) L-(+)-arabinose.

The H7-Fc and H9-Fc bivalent constructs as well as the tetravalent, biparatopic H7-H9-fFc molecule were designed in silico and synthesized (Biocat). Regarding the bivalent constructs, the Nbs (H7 and H9) were fused to the hinge region and Fc domain of the subclass B of canine IgG[83]. Concerning the tetrameric biparatopic molecule, H7-H9 was further fused to the hinge region and Fc domain of the ferret IgG, including the S252Y substitution reported to improve half-life[67]. All Fc-carrying constructs were produced at the protein-expression core facility of the École Polytechnique Fédérale de Lausanne (EPFL, Switzerland; Protein Production and Structure Core Facility) as described above for solH expression. Subsequently, the proteins were purified from the supernatant by using a 5 mL HiTrap rProtein A FF column (Cytiva) with the following buffer: 100 mM Tris-HCl pH 8.0, 150 mM NaCl, and 1 mM EDTA. Proteins were finally eluted using 100 mM sodium citrate pH 3.2 and rebuffered into PBS using PD-10 Desalting Columns (Cytiva) following the manufacturer's instructions.

### Immunofluorescence analysis

Vero cells (60,000 cells/well) were seeded in 24-well plates (Corning) and transfected with 1 μg per well of a plasmid encoding the Flag-tagged CDV H (strain A75/17) protein[84]. 24 h post transfection, unfixed and unpermeabilized cells were incubated with 10 μg/ml of c-Myc-tagged Nbs in Opti-MEM (Gibco) (1 h at 4 °C). Subsequently, the cells were fixed with 4% paraformaldehyde (PFA). Treated cells were incubated with a monoclonal mouse anti-c-Myc antibody (Invitrogen; 1:500 dilution in PBS) for 1 h at 4 °C. Next, cells were decorated with an Alexa Fluor 488-conjugated goat anti-mouse IgG (H+L) (Invitrogen; 1:1000 dilution in PBS) and DAPI (Thermo Scientific; 1:1000 dilution in PBS) for 1 h at 4 °C. Pictures were acquired by laser scanning confocal microscopy by using an Olympus Fluoview FV3000 Confocal Laser Scanning Microscope and were then analyzed and composed using the Fiji software.

### Surface plasmon resonance

The SPR experiments were performed using OpenSPR (Nicoya) equipped with a carboxyl sensor. The ligand (solH) was covalently immobilized using amine-coupling chemistry. The surfaces of two flow cells were conditioned with 10 mM HCl at a flow rate of 150 μl/min. Subsequently, the two flow cells were activated for 5 min with a 1:1 mixture of 0.1 M NHS (N-hydroxysuccinimide) and 0.1 M EDC (3-(N,N-dimethyl amino) propyl-N-ethylcarbodiimide) at a flow rate of 20 μl/min. The ligand at a concentration of 25 μg/ml in SPR immobilization buffer (10 mM Na-acetate pH 4.5 from IBA-Lifesciences) was immobilized on the flow cell 2 for 5 min; flow cell 1 was left blank to serve as a reference surface. Both surfaces were blocked with a 5 min injection of 1 M ethanolamine-HCl pH 8.5 at a flow rate of 20 μl/min. To collect the kinetic binding data, the analyte (Nbs or Nb constructs) in HNE-T buffer (100 mM HEPES-NaOH pH 7.4, 150 mM NaCl, 3.4 mM EDTA, 0.05% (v/v) Tween 20) was injected over the two flow cells at increasing concentrations at a flow rate of 20 μl/min and a temperature of 20 °C. The complex was allowed to associate and dissociate for 90 and 500 s, respectively. The surfaces were regenerated after every injection with regeneration buffer (3 M guanidinium-HCl from IBA-Lifesciences) at a flow rate of 150 μl/min. Duplicate injections of each sample were performed and the data were fit to a 1:1 binding model within TraceDrawer software.

For epitope binning experiments, solH was captured as described above. Then, Nb H7 was injected at a saturating concentration of 250 nM, followed by another injection of Nb H7 at 250 nM premixed with Nb H7, Nb H9, or an irrelevant Nb at a concentration of 100 nM. Identical experiments were repeated starting with Nb H9.

### Competition ELISA

For the competition ELISA, MaxiSorp™ 96 well plates (Nunc) were coated with 30 nM of purified Nbs carrying a C-terminal myc-His tags (4 °C, overnight). In parallel, 0.5 μg/ml TST-tagged solH protein was incubated with 600 nM Nbs in PBS-BSA-T buffer (PBS containing 0.5% (w/v) bovine serum albumin (BSA) and 0.05% (v/v) Tween-20 (T)) (4 °C, overnight). The next day, wells were blocked with PBS-BSA-T (1 h at room temperature (RT)). Pre-mixed solH/Nb complexes were then supplemented to the coated wells (1 h at RT). Wells were then incubated with the NWSHPQFEK Tag Monoclonal Antibody (GenScript; 1:5000 in PBS-BSA-T) (1 h at RT). Finally, the polyclonal Goat Anti-Mouse Immunoglobulins/HRP was added (Agilent; 1:3000 dilution in PBS-BSA-T). Detection was performed using TMB Substrate Solution (Thermo Scientific) (5 min at RT). Subsequently, the reaction was stopped by the addition of 1 M sulphuric acid, and the absorbance at 450 nm was measured with the cell imaging multi-mode reader Cytation 5 (BioTek).

For the SLAM/Nb competition ELISA, 1 μg/ml of solH was coated in MaxiSorp™ 96 well plates (Nunc) (4 °C, overnight). Nbs diluted in PBS-T were then added to the wells (starting concentration of 30 μg/ml for the monovalent Nbs and 10 μg/ml for the biparatopic Nb) (1 h at RT). To obtained $IC_{50}$ values, serial dilutions (1/10) of the Nbs were performed. As negative control, solH without addition of any Nb was also

included. Wells were further treated with soluble V-domain canine SLAM receptor (solSLAM.V-Fc[64]; 1 μg/ml in PBS-T) (1 h at RT). Detection was performed using a goat anti-human IgG Fc/HRP (Merck Millipore; 1:10,000 dilution in PBS-T) (1 h at RT). Detection was performed using TMB Substrate Solution (Thermo Scientific) (5 min at RT). The reaction was stopped by the addition of 1 M sulphuric acid, and the absorbance at 450 nm was measured with the cell imaging multi-mode reader Cytation 5 (BioTek).

## Titration ELISA

For the titration ELISA, MaxiSorp™ 96 well plates (Nunc) were coated with 100 ng solH (CDV or MeV) protein carrying the TST-Tag (4 °C, overnight). The next day, the samples were serially diluted 1:10 in PBS-T, starting with 1 μM and added to the washed wells of the ELISA plate (RT, 3 h). Wells were then incubated with the c-Myc Epitope Tag Mouse Monoclonal Antibody (Invitrogen; 1:2000 in PBS-T) (2 h at RT). Finally, the polyclonal Goat Anti-Mouse Immunoglobulins/HRP was added (Agilent; 1:3000 dilution in PBS-T). Detection was performed using TMB Substrate Solution (Thermo Scientific) (5 min at RT). Subsequently, the reaction was stopped by the addition of 1 M sulphuric acid, and the absorbance at 450 nm was measured with the cell imaging multi-mode reader Cytation 5 (BioTek).

## Neutralization assays

Cell entry-inhibition assay: Vero-cSLAM (25,000 cells/well) cells were seeded in 96-well plates (polystyrene microplates with a microclear bottom; Greiner Bio One) and incubated at 37 °C for 2–3 h. In parallel, Nbs and Nb constructs were serially diluted (1/10) in DMEM and pre-incubated with the viral construct A75/17[neon/nlucP] in 100 μl for 30 min at 37 °C. As negative controls, viruses without the addition of any Nb (or Nb constructs) were treated in a similar manner. The Nb/virus mixtures were then added to Vero-cSLAM cells (multiplicity of infection (MOI) of 0.03) and the cells were further incubated at 37 °C. 24 h post infection, luminescence signals were recorded using the Cytation 5 device (BioTek). Briefly, the NanoGlo® Live Cell Substrate (Promega) was diluted 1:20 in NanoGlo® LCS Dilution Buffer (Promega), followed by a further dilution of 1:10 in Opti-MEM.

Progeny virus-inhibition assay: Vero-cSLAM cells (500,000 cells/well) were seeded in 6-well plates and incubated at 37 °C for 24 h. H7-H9-fFc was serially diluted (1/10) in DMEM and preincubated with either CDV-5804P-GFP[78], CDV-A75/17[85], or CDV-OL-GFP in 2 ml for 1 h at 4 °C. As negative controls, viruses without the addition of H7-H9-fFc were treated in a similar manner. The mixtures were then added to Vero-cSLAM cells (MOI of 0.01) and the cells were further incubated at 37 °C. 24 h post infection (48 h for CDV-OL-GFP), cell-associated viruses were harvested in 100 μl OptiMEM (Thermo Fisher Scientific). Samples were subjected to three freeze-thaw cycles and subsequently the cell debris were removed by centrifugation at $300 \times g$. Virus titers were determined on Vero-cSLAM cells using the 50% tissue culture infectious dose (TCID$_{50}$) method.

## Cryo-EM sample and grid preparation

For cryo-EM sample preparation, the N-terminal affinity cassette composed of a His- and Twin-Strep-tag was cleaved off from purified CDV-solH (see section 'Expression and purification of CDV H protein ectodomain') using human rhinovirus 3C protease (HRV3C, BioVision) at a two-molar excess and in the presence of 0.25% (w/v) octyl-β-glucoside (OG) for 2 h at RT. HRV3C cleaved CDV-solH was concentrated using an Amicon 100-kDa molecular weight cut-off (MWCO) device (Millipore) and further purified by size-exclusion chromatography (SEC). Preparative SEC was conducted at 8 °C on an Äkta Pure system (Cytivia) using a Superose 6 10/300 GL column (Cytivia) and a running buffer consisting of 20 mM Tris-HCl pH 7.6, 100 mM NaCl, 0.25% OG (w/v). Peak fractions corresponding to CDV-solH tetramers were pooled, concentrated to about 3 mg/ml using an Amicon 100-kDa

MWCO concentrator and centrifuged for 15 min at $100,000 \times g$ and 4 °C. For complex formation, the HRV3C-cleaved and purified CDV-solH protein, and Nbs H7 and Nb H9 were incubated at 4 °C for 1 h (CDV-solH/Nb H7/Nb H9 molar ratio = 1/2.5/2.5) at a final protein concentration of 1.2 mg/ml. The mixture was centrifuged for 5 min at $20,000 \times g$ and 4 °C right before grid preparation. Then, 3 μl of CDV-solH/Nb H7/Nb H9-complex was applied onto glow-discharged (15 s at 10 mA, and 0.25 mbar) holey carbon grids (R 2/1, Au 200 mesh, Quantifoil Micro Tools GmbH, Germany). Samples were blotted for 4.5 s at 4 °C and 100% humidity and vitrified in liquid ethane using a Vitrobot Mark IV system (Thermo Fisher).

## Cryo-EM data acquisition

Cryo-EM data were collected on a Titan Krios G4 cryo-transmission electron microscope (Thermo Fischer Scientific) operated at an acceleration voltage of 300 kV and equipped with a Direct Electron Detector (DED) CMOS Falcon 4i camera with Selectris energy filter (slit width 20 eV), operating in electron event representation (EER) mode[86]. The Falcon 4i camera was calibrated at a nominal magnification of 165,000× resulting to a calibrated pixel size of 0.733 Å at the specimen level. The nominal defocus used to collect data ranged from Δz = −0.8 to −2.5 μm. A total of 9162 movies were collected with the software smart EPU (Thermo Fischer Scientific). The camera was set up to collect 666 raw EER frames in counting mode with a total exposure time of 2.2 s, resulting in a total dose equivalent to 30 electrons/Å$^2$ per exposure.

## Image processing and single particle analysis

Collected movies were compressed to TIFF format (EER-frames grouped to get about 1 e⁻/Å/frame) with the "relion_convert_to_tiff" tool of Relion (version 4)[87]. Beam-induced motion correction of dose-fractionated and gain-corrected TIFF compressed movies were performed using MotionCor2 (version 1.4)[88] with a 5×5 patch-based alignment. Contrast transfer function (CTF) estimation of full-frame non-weighted micrographs was carried out with Gctf (version 1.6)[89] as a standalone program. 3977 micrographs of lower quality displaying strong drift, astigmatism >500 Å, and CTF resolution >5 Å were excluded from further processing. Subsequently, a non-template-driven, convolutional neural network-based particle picking was performed within crYOLO (version 1.5)[90]. A total of 304,110 particles were auto-picked using with the pre-trained general model and extracted from the 5185-remaining dose-weighted micrographs with Relion (version 4). Multiple rounds of two-dimensional (2D) classification were then carried out in Relion. The resulting 2D class averages of the CDV-solH/Nbs-complex from a total of 286,817 particles were used for ab initio reconstruction with cryoSPARC (version 4.2)[91] and 3D classification with C$_1$ symmetry. The particles from similar 3D classes were merged to be used for future refinement to 4.3 Å resolution in cryoSPARC. From this map, individual masks for each dimer of the CDV-solH tetramer were generated for subsequent local refinement. The binary masks were created with the "relion_mask_create" tool using at least three pixels extension and ten pixels for smoothening the edges. As estimated by Fourier Shell Correlation (FSC)[92] calculation with the 0.143 cut-off criterion, final resolutions of 3.1 Å (dimer I) and 3.4 Å (dimer II) were obtained (Supplementary Fig. 2). Local resolution estimation was done with the tool MonoRes[93] implemented in cryoSPARC (Supplementary Fig. 3).

## Atomic model building and refinement

The model of the previously solved CDV-solH tetramer structure[12] was positioned into the 4.3 Å reference cryo-EM map using Chimera X (version 1.7.1)[94]. Coordinates of the individual CDV-solH dimers were then refined against their corresponding locally refined density maps. Subsequently, AlphaFold2[95] generated models of Nbs H7 and H9 were oriented and fitted into the corresponding Nb densities based on their

amino acid sequence and location of disulfide bridges. The individual CDV-solH/Nbs dimer complex models were obtained by several iterative rounds of manual model building in Coot (version 0.9.8.5)[96] and real-space refinement in Phenix (version 1.20)[97]. The coordinates of dimer I and II were combined into a single CDV-solH tetrameric Nbs-bound structure for the final validation using MolProbity[98]. Data collection, model refinement and validation statistics are summarized in Supplementary Data 6 and analysis of the final cryo-EM density map is shown in Supplementary Fig. 3. The final model and cryo-EM maps were deposited in the Protein Data Bank (PDB; accession code: 9HBP) and the Electron Microscopy Data Bank (EMDB; accession code EMD-52020, reference map; EMD-52024, composite map; EMD-52023, locally refined map for dimer I at 3.1 Å resolution and EMD-52021, locally refined map for dimer II at 3.4 Å resolution).

## Qualitative fusion assays

Vero-cSLAM cells were seeded in 6-well plates (Corning) (500,000 cells/well). On the next day, the cells were transfected with plasmids encoding for CDV H (strain A75/17) protein, CDV F (strain A75/17) protein, and GFP (1, 2, and 0.5 µg per well, respectively). 4 h post transfection, 5 µg/ml of Nbs were added to the transfected cells. Images were acquired 72 h post transfection with an EVOS M5000 fluorescence microscope (10× objective). In parallel, Vero-cSLAM cells were seeded in 6-well plates (Corning) (500,000 cells/well) and incubated at 37 °C for 2–3 h. Subsequently, cells were infected with the viral construct A75/17$^{neon/nlucP}$ (MOI of 0.1). 4 h post infection, 5 µg/ml of Nbs were added to the infected cells. Images were acquired 2–3 days post infection with an EVOS M5000 fluorescence microscope (4× objective).

## Quantitative fusion assays

Quantitative fusion assays were performed as previously described[43] with some modifications. Vero cells (effector cells) were seeded in 96-well plate (polystyrene microplates with a microclear bottom; Greiner Bio One) (30,000 cells/well) and Vero-cSLAM cells (target cells) were seeded in 6-well plates (Corning) (500,000 cells/well). 6 h later, Vero cells were transfected with plasmids encoding for CDV H (strain A75/17) protein or derivative mutants, CDV F (strain A75/17) protein, and RFP-HiBiT (0.066, 0.133, 0.1 µg per well, respectively). On the other hand, Vero-cSLAM cells were transfected with 3 µg per well of a plasmid encoding for GFP-LgBiT. 24 h post transfection, Nbs were serially diluted (at the indicated dilution) in DMEM and added to the effector cells. After incubation at 37 °C for 30 min, the target cells were trypsinized and supplemented to the effector cells. 18 h post cell mixing, luminescence signals were recorded using the Cytation 5 device (Bio-Tek). Briefly, the NanoGlo® Live Cell Substrate (Promega) was diluted 1:20 in NanoGlo® LCS Dilution Buffer (Promega), followed by a further dilution of 1:10 in Opti-MEM.

## Ferret experiments

Groups of three ferrets (*Mustela putorius furo*) aged 1 year or older, consisting of two females and one male, were infected intranasally with $2 \times 10^5$ TCID$_{50}$ of recombinant CDV strain 5804P, which expresses GFP from an additional gene cassette between the H and L genes (5804P-GFP)[78]. One group of ferrets was treated with 1 mg/kg body weight of the biparatopic H7-H9-fFc construct by intraperitoneal injection starting 3 days before infection (4×H7-H9-fFc treatment group), while the other group received the first treatment on the day of infection (3×H7-H9-fFc treatment group). Both groups were subsequently treated on days 3 and 7 after infection. Two groups of three ferrets each were only treated after infection. One group received the same dose of H7-H9-fFc (1 mg/kg body weight) on days 3 and 7 (2×H7-H9-fFc treatment group), while a final group of three ferrets were treated only on day 7 (1×H7-H9-fFc treatment group). As a control, one group of three ferrets (two females, one male) was infected intranasally with $2 \times 10^5$

TCID$_{50}$ of CDV 5804P-GFP. These animals were not given any antibody treatments.

Animals were monitored daily for signs of clinical disease (i.e., fever, rash, and weight loss), and their general condition was evaluated[99]. Blood was sampled twice weekly under general anesthesia, where white blood cell counts were performed and cell-associated virus titers were determined from purified PBMCs and expressed as TCID$_{50}$ per $10^6$ PBMCs[100]. When pre-defined clinical endpoints were reached (temperature >40 °C for three consecutive days or >41 °C; weight loss >15% of initial study weight), the respective animals were euthanized under general anesthesia and exsanguinated, followed by post-mortem sampling of organs.

## Analysis of RNA from ferret tissues

Organ samples were homogenized in lysing matrix (MP Biomedicals) with DMEM (Sigma) without FBS or directly in TRIzol (Invitrogen). cDNA was generated from total RNA by reverse transcription using SuperScript IV RTase (Invitrogen) and random hexamers (IDT). The H gene was amplified using gene-specific primers, and amplicons were sequenced using the Sanger method.

## Statistics and reproducibility

Statistical analyzes were performed using GraphPad Prism software (version 9 and version 10) and TraceDraver (version 1.9.2). In neutralization assays, half-maximal neutralization concentrations were determined from serial dilution curves generated in GraphPad Prism and fitted with a four-parameter curve. In the MeV H protein-based ELISAs, curves were generated in GraphPad Prism and fitted with a three-parameter curve. For all other ELISAs, graphs were generated with GraphPad Prism and fitted with a four-parameter curve. Binding data from openSPR were fitted with a 1:1 binding model (TraceDrawer). For ferret experiments, groups are shown as the mean values of three animals ($n = 3$) with standard deviations indicated by error bars, unless otherwise indicated. Animals were randomly assigned to groups, each which were comprised of 1 male and 2 females, based on availability from the Paul-Ehrlich-Institute ferret breeding facility. Animals from the same litter were given preference for group assignment in order to reduce conflict and stress during housing. Castrated males were used due to their reduced levels of aggression. All animals were at least 1 year old at the beginning of the studies. Animal support staff and caretakers were blinded, but investigators were not due to limitations in the number of personnel who are authorized and trained to perform ferret studies. No data were excluded from the analyzes.

## Reporting summary

Further information on research design is available in the Nature Portfolio Reporting Summary linked to this article.

## Data availability

Protein model coordinates and cryo-EM maps were deposited in the Protein Data Bank (PDB); accession code 9HBP and the Electron Microscopy Data Bank (EMDB); accession code EMD-52020, reference map; EMD-52024, composite map; EMD-52023, locally refined map for dimer I at 3.1 Å resolution and EMD-52021, locally refined map for dimer II at 3.4 Å resolution. All data associated with this study are presented in the paper or the Supplementary Information. Source data are provided with this paper.

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

## Acknowledgements

Cryo-EM analyses were performed by employing the equipment supported by the Microscopy Imaging Center (MIC), University of Bern, Switzerland and Dubochet Center for Imaging (DCI) Bern, Switzerland. This work was funded by the University of Bern and the Swiss National Science Foundation (SNSF; CRSII5_183481 to R.R., P.P., and D.F., and SNSF 310030_204363 to P.P.) and the Deutsche Forschungsgemeinschaft (DFG; CRC 1021 Project 197785619/B12 to C.K.P. and B.S.).

## Author contributions

P.P., M.S., D.F., and B.S. conceived the project. M.S., N.D., O.S., J.-M.J., M.W., M.C., and M.D.P.S. performed experiments. P.P., J.-M.J., D.F., P.C., and B.S. supervised research. P.P., D.F., R.R., P.C., C.K.P., and B.S. obtained funding. M.S., B.S., D.F., and P.P. wrote the first drafts of the paper. All authors reviewed and approved the final manuscript.

## Competing interests

M.S., P.C., B.S., D.F., and P.P. are listed as inventors on a patent application that is related to the subject matter of this manuscript. All other authors declare to have no competing interests regarding the work described in this manuscript.
