## [Transparent Peer Review file · Nature Communications]

Protection against lethal canine distemper virus infection by a dual epitope-targeting synthetic antibody

Corresponding Author: Professor Philippe Plattet

Version 0:

Reviewer comments:

Reviewer #1

(Remarks to the Author)

This manuscript investigates a novel dual epitope-targeting synthetic antibody aimed at providing protection against lethal morbillivirus infections, with a focus on canine distemper virus (CDV). The study identifies two potent nanobodies, H7 and H9, targeting distinct epitopes on the tetrameric H protein of CDV. The epitopes and mechanisms of inhibition are investigated through functional, biochemical, biophysical, and cryo-EM analyses.

The authors then engineered a biparatopic construct (H7-H9) that displays significantly enhanced neutralization profiles, surpassing the activity of each monovalent Nbs. The authors demonstrate the prophylactic and therapeutic efficacy of an engineered tetravalent biparatopic antibody (H7-H9-fFc) in a ferret model of CDV infection. The research concludes that the engineered antibody, which targets two distinct viral antigenic sites simultaneously and neutralizes through different molecular mechanisms, was an effective antiviral against CDV in ferrets. The work is innovative, showcasing a combination of molecular biology, structural biology, and in vivo experimentation to design a therapeutic antibody format with enhanced potency. Although CDV can be largely ameliorated by available live attenuated vaccines, the authors note that some susceptible and endangered species (e.g. pandas) are prone to develop disease after vaccination.

The manuscript contributes to the broader field of antiviral therapeutics by focusing on morbillivirus infections, particularly those cases where vaccines may not be sufficient. The structural analysis provided through cryo-EM is thorough and offers significant insights into epitope targeting, enriching our understanding of antibody-antigen interactions. The in vivo ferret model is robustly designed, convincingly demonstrating the therapeutic potential of the engineered biparatopic antibody. Furthermore, the innovative biparatopic antibody design stands out as a promising approach, highlighting the potential for combining structural and functional mechanisms to enhance neutralization potency. These strengths collectively underscore the manuscript's contribution to advancing therapeutic strategies against paramyxoviruses.

I have few comments that will further improve the clarity or strengthen the impact of this study for the morbillivirus field:

1. MeV and CDV induced immune amnesia is a well-established phenomenon and can contribute to the morbidity associated with morbillivirus infections. The authors should test if the Nb treatment also prevents immune amnesia. My understanding that laboratory ferrets are always immunized against rabies, so the authors could test if titers of anti-rabies antibodies are affected in the various treatment groups.
2. Fig. 5d, I would plot this in a similar manner to Fig. 5f so that the difference in viral titers between the groups can be more easily visualized. As currently rendered, the line graph in Fig. 5d makes it difficult to assess the differences in viremia between the various groups especially on day 3, day 7 and 10.
3. Extended Data Fig. 6f – why is the OL vaccine strain neutralized so much less (~ 2 logs) than the other two strains used? Are there mutations in the epitope of Nb H7 and/or H9? Please clarify.

Minor comments:

- Extended Data Fig. 1a legend: Clarify the specific antibodies used in immunofluorescence experiments. Although it appears that Nb H7, Nb H9 and H7-H9 were used, the legend also indicates that an anti-c-Myc antibody was used? This doesn't make sense.
- Line 407: "infections in captive animals is a recurring problem" should be "infections in captive animals are a recurring problem."

Reviewer #2

(Remarks to the Author)

The manuscript entitled "Protection against lethal morbillivirus infection by a dual epitope-targeting synthetic antibody" by Melanie Scherer et al. provides a comprehensive examination of two CDV-specific nanobodies, H7 and H9, targeting two neutralizing sites on the CDV H protein. The authors further constructed an engineered tetravalent and bivalent antibody, H7-H9-fFc, that displayed improved neutralization capacity. Additionally, H7-H9-fFc showed protective efficacy in ferrets, effectively preventing fatal viral infections at low doses, thus offering a promising therapeutic approach against CDV infections. However, several major concerns need to be addressed before considering future applications:

1. The title of this study claims the protection of synthetic antibody against lethal morbillivirus infection. However, the results of antibody characterizations are limited exclusively to CDV. This emphasis on CDV restricts the broader impact of the findings. Actually, as noted in the Discussion section, "the receptor-binding H protein of MeV shows only 35% sequence identity compared to that of the CDV, and multiple residues vary within both neutralizing epitopes (Supplementary Tables 5)." The cross-neutralization activity of the antibodies is an important criterion for evaluating the significance of this study. Since CDV infections may not necessitate antibody therapeutics since existing vaccines are already effective in controlling viral effects. Given MeV's human health relevance, the authors should include necessary validation of the cross-reactivity of H7, H9 and H7-H9-fFc with MeV H protein via binding and/or neutralization assays. Additionally, a discussion on the structural conservation of sites I/II among morbillivirus species should be included.
2. This study also lacks a comprehensive evaluation of the broad-neutralizing capabilities of Nbs and the synthetic antibody against the mutant CDV strains. Although the author claim that the epitopes of H7 and H9 are highly conserved among different CDV strains, many mutations (e.g. A544 site of the H7 epitope) were found in the epitopes of both antibodies. This raises concerns about the broad-spectrum activities of these antibodies. Consequently, the author should perform structural analysis or experiments to assess the impact of these mutations on of both antibodies.
3. This manuscript lacks a necessary discussion on the limitations of current CDV vaccines and does not provide a comprehensive review of existing research on CDV H protein-targeted drug, including studies on morbillivirus neutralizing antibodies. This omission conceals the need for developing antibody therapies. The authors should include a detailed summary of these topics in the Introduction section.
4. While Nb H7's receptor competition mechanism is well characterized, the precise mode of neutralization by Nb H9 remains speculative. The authors hypothesize that Nb H9 might disrupt H/F conformational coupling or F triggering. However, no any evidence were provided. The author should provide more evidence (e.g. compare H-F complex stability with/without Nb H9) to addressing the mechanistic elucidation of Nb H9.
5. The risk of viral resistance is a critical concern in antibody drug development. In the animal assays conducted in this study, while most ferrets administered with multiple H7-H9-fFc injections were protected against a lethal CDV challenge. However, the delayed viral clearance and death of one (1/3) 4x-dose-treated ferret indicates that the potential emergence of resistant mutants. An S546Y mutant was indeed isolated from the administrated animal. Although the authors demonstrated that the antibody could neutralize this mutant strain in vitro, however, this evidence alone did not support the conclusion that "No drug escape variant emerged upon treatment in vivo". Furthermore, the structural analysis of S546Y's impact on H7 binding appears unreasonable. The authors need to perform in vitro virus resistance selection experiments using H7, H9 and H7-H9-fFc respectively, to identify potential mutations impacting antibody binding, and compare escape mutation frequencies between dual-epitope and single-epitope antibodies. Additionally, the authors should also conduct an additional animal assay by challenging ferrets with the S546Y mutant to evaluate the efficacy of H7-H9-fFc.

Minor comments:

1. the parameters for Poor Rotamer (1.37) and All-atom clashscore (13.04) are relatively high, the structural model may require further refinement.
2. The structural presentation in Figure 2 are confusing, please provide at least the entire cryo-EM density map color by elements and a dimeric model representation including one CDV-solH dimer bound by one H9 and two H7.
3. Clarify statistical methods, e.g., error bars in Fig. 5c,d; n-values for replicates.
4. Please include the ethical clearance for this study.

Version 1:

Reviewer comments:

Reviewer #1

(Remarks to the Author)

The authors have satisfactorily answered my critiques.

Reviewer #2

(Remarks to the Author)

The authors have addressed some of the previous concerns, however, several issues remain either unresolved or insufficiently addressed. Below are the major and minor points that still need attention.

Comment 1: In the rebuttal, although the authors provided additional ELISA data (not functional data), it remains insufficient as it does not fully address the core issue of cross-neutralization activity. The absence of neutralization assays against MeV leaves the key question unanswered, specifically whether the antibody has broader applicability to the Morbillivirus genus. Moreover, the weak binding efficiency to MeV H protein, as demonstrated in the additional ELISA, underscores that the study's scope is functionally limited to CDV, thereby reinforcing that the current title overstates the findings. While the CDV ferret model is well-established, it primarily validates an intervention against CDV itself. To legitimately claim genus-level

protection, direct evidence, or at least cross-neutralization data against other morbilliviruses, particularly MeV, is essential, which the authors have not provided.

Comment 2: I commend the authors for utilizing quantitative membrane fusion assays to assess the broad-neutralizing capabilities of Nb H7, H9, and H7-H9 against mutant CDV strains. The data presented are compelling, demonstrating that the H7-H9 construct maintains efficacy against single mutants, thus strengthening the manuscript. However, at a concentration of 0.5 µg/ml, the dual mutant (G324W/A325G) significantly challenges the potency of H7-H9, suggesting reduced effectiveness. This finding implies that naturally occurring combinatorial mutations in circulating strains may compromise the H7-H9's broad-neutralizing capability. I recommend that the authors explicitly discuss this limitation in the manuscript to provide a comprehensive interpretation of the effectiveness of H7-H9.

Comment 3: Addressed.

Comment 4: I acknowledge the authors' efforts in attempting co-immunoprecipitation experiments to elucidate the neutralization mechanism of Nb H9. However, as the assay yielded inconclusive results, the proposed mechanism remains speculative. The authors need to state in the text that the mechanism of Nb H9 neutralization remains undefined due to a lack of experimental evidence, ensuring conclusions are appropriately cautious.

Comment 5: In rebuttal, the authors demonstrated the use of quantitative membrane fusion assays to evaluate the broad-neutralizing capabilities of Nbs. However, the S546Y mutant, which was isolated from animal experiments and has significant practical implications, was not included in the in vitro evaluation. Considering that the authors did not intend to conduct the screening of escape mutants that I suggested, to thoroughly assess the antibody's drug resistance risk, it is recommended to include fusion inhibition assay data for the S546Y mutant. This inclusion would strengthen and substantiate the argument for the broad-neutralizing capability of H7-H9.

Minor comments: Addressed.

REVIEWER COMMENTS

Reviewer #1 (Remarks to the Author):

This manuscript investigates a novel dual epitope-targeting synthetic antibody aimed at providing protection against lethal morbillivirus infections, with a focus on canine distemper virus (CDV). The study identifies two potent nanobodies, H7 and H9, targeting distinct epitopes on the tetrameric H protein of CDV. The epitopes and mechanisms of inhibition are investigated through functional, biochemical, biophysical, and cryo-EM analyses.

The authors then engineered a biparatopic construct (H7-H9) that displays significantly enhanced neutralization profiles, surpassing the activity of each monovalent Nbs. The authors demonstrate the prophylactic and therapeutic efficacy of an engineered tetravalent biparatopic antibody (H7-H9-fFc) in a ferret model of CDV infection. The research concludes that the engineered antibody, which targets two distinct viral antigenic sites simultaneously and neutralizes through different molecular mechanisms, was an effective antiviral against CDV in ferrets. The work is innovative, showcasing a combination of molecular biology, structural biology, and in vivo experimentation to design a therapeutic antibody format with enhanced potency. Although CDV can be largely ameliorated by available live attenuated vaccines, the authors note that some susceptible and endangered species (e.g. pandas) are prone to develop disease after vaccination.

The manuscript contributes to the broader field of antiviral therapeutics by focusing on morbillivirus infections, particularly those cases where vaccines may not be sufficient. The structural analysis provided through cryo-EM is thorough and offers significant insights into epitope targeting, enriching our understanding of antibody-antigen interactions. The in vivo ferret model is robustly designed, convincingly demonstrating the therapeutic potential of the engineered biparatopic antibody. Furthermore, the innovative biparatopic antibody design stands out as a promising approach, highlighting the potential for combining structural and functional mechanisms to enhance neutralization potency. These strengths collectively underscore the manuscript's contribution to advancing therapeutic strategies against paramyxoviruses.

Authors: We would like to thank Reviewer #1 for the positive assessment of our work.

I have few comments that will further improve the clarity or strengthen the impact of this study for the morbillivirus field:

1. MeV and CDV induced immune amnesia is a well-established phenomenon and can contribute to the morbidity associated with morbillivirus infections. The authors should test if the Nb treatment also prevents immune amnesia. My understanding that laboratory ferrets are always immunized against

rabies, so the authors could test if titers of anti-rabies antibodies are affected in the various treatment groups.

Authors: Although determining the influence of our drug regarding CDV-induced immune amnesia is a very interesting topic, in our study, the ferrets were not vaccinated against rabies prior to our experiments. We therefore decided to keep the focus of the manuscript exclusively on the therapeutic potential of our drug against CDV.

2. Fig. 5d, I would plot this in a similar manner to Fig. 5f so that the difference in viral titers between the groups can be more easily visualized. As currently rendered, the line graph in Fig. 5d makes it difficult to assess the differences in viremia between the various groups especially on day 3, day 7 and 10.

Authors: As suggested by the Reviewer, we revised Figure 5d to show viremia by groups (including error bars). Differences between the various group treatments can now be more clearly visualized.

3. Extended Data Fig. 6f – why is the OL vaccine strain neutralized so much less (~ 2 logs) than the other two strains used? Are there mutations in the epitope of Nb H7 and/or H9? Please clarify.

Authors: This is a very interesting question, which was also raised by points #2 and #5 of Reviewer #2 (see also our answers below). Briefly, we conducted additional experiments to identify which amino acid residues locating within the Nb H7 and H9 epitopes and different in selected circulating CDV strains, may affect their potency. Strikingly, mutations H protein-M500I and H protein-G324W strongly interfered with Nb H7 and H9 efficacy, respectively (shown in a revised Supplementary Table 7). Notably, I500 and W324 are present in the H protein of the OL vaccine strain. A new paragraph describing these results was added in the revised version of the manuscript (pages 8-9; lines 261-285).

Minor comments:

- **Extended Data Fig. 1a legend: Clarify the specific antibodies used in immunofluorescence experiments. Although it appears that Nb H7, Nb H9 and H7-H9 were used, the legend also indicates that an anti-c-Myc antibody was used? This doesn't make sense.**

Authors: As mentioned in the Material & Methods section (Immunofluorescence analysis), both monovalent Nbs as well as the biparatopic construct are his- and c-myc-tagged. We thus treated H-expressing cells initially with the various Nbs, followed by the addition of an anti-c-myc mAb and an Alexa fluor-conjugated secondary antibody for detection. We modified the sentence accordingly in the legend of Extended Data Figure 1a (page 41, line 1166).

- **Line 407: "infections in captive animals is a recurring problem" should be "infections in captive animals are a recurring problem."**

Authors: Thank you, we have modified accordingly (page 13, line 440).

Reviewer #2 (Remarks to the Author):

The manuscript entitled "Protection against lethal morbillivirus infection by a dual epitope-targeting synthetic antibody" by Melanie Scherer et al. provides a comprehensive examination of two CDV-specific nanobodies, H7 and H9, targeting two neutralizing sites on the CDV H protein. The authors further constructed an engineered tetravalent and biparatopic antibody, H7-H9-fFc, that displayed improved neutralization capacity. Additionally, H7-H9-fFc showed protective efficacy in ferrets, effectively preventing fatal viral infections at low doses, thus offering a promising therapeutic approach against CDV infections.

Authors: We thank Reviewer #2 for the careful and thorough assessment of our work and for supporting our hypothesis that multivalent and multiparatopic nanobody-based antibodies could offer clinical advantages in the future management of viral infections.

However, several major concerns need to be addressed before considering future applications:

1.1 The title of this study claims the protection of synthetic antibody against lethal morbillivirus infection. However, the results of antibody characterizations are limited exclusively to CDV. This emphasis on CDV restricts the broader impact of the findings. Actually, as noted in the Discussion section, "the receptor-binding H protein of MeV shows only 35% sequence identity compared to that of the CDV, and multiple residues vary within both neutralizing epitopes (Supplementary Tables 5)." The cross-neutralization activity of the antibodies is an important criterion for evaluating the significance of this study. Since CDV infections may not necessitate antibody therapeutics since existing vaccines are already effective in controlling viral effects. Given MeV's human health relevance, the authors should include necessary validation of the cross-reactivity of H7, H9 and H7-H9-fFc with MeV H protein via binding and/or neutralization assays.

Authors: We fully agree with the Reviewer's point of view that the development of antiviral drugs against measles virus (MeV) is timely, since the availability of authorized drugs may be key to support the global measles eradication program, help to control epidemics and provide an alternative to immunocompromised patients that cannot be vaccinated.

While our long-term aim is indeed to design next-generation therapeutics against MeV, the primary goal of the present study was to provide proof-of-principle for a nanobody-based antiviral strategy to treat morbillivirus infections. In this context, CDV-infected ferrets represent a well-established and cost-effective animal model to evaluate the potency of novel antiviral approaches against morbilliviruses and other viruses (e.g., Therapeutic mitigation of measles-like immune amnesia and exacerbated disease after prior respiratory virus infections in ferrets, 2024. *Nat Commun*. DOI: 10.1038/s41467-024-45418-5). We modified the Introduction section to provide more information on the model system employed (page 4; lines 112-114).

While we fully agree with the Reviewer that a single antiviral drug capable of blocking both CDV and MeV infections would be highly valuable for human and veterinary medicine, the relatively low amino acid sequence identity between the MeV and CDV H proteins (about 35%) does not favor such a drug profile. Nevertheless, and as requested, we have now provided additional ELISA data showing that Nb H7 and H9 bind only weakly to MeV H (revised Extended Data Fig. 1d). Although the biparatopic H7-H9 construct exhibited somewhat improved binding to MeV H compared to the monovalent format, its binding efficiency remained about 1000-fold lower than that observed for CDV H. These findings explain why we did not pursue further experiments with MeV H. Corresponding clarifications have been added to the revised manuscript (page 4; lines 127-130 and page 8; lines 252-255).

1.2. Additionally, a discussion on the structural conservation of sites I/II among morbillivirus species should be included.

Authors: We have revised Supplementary Tables 2 and 3 to include the amino acids of MeV H (wt-ICB and vaccine-Edmonston strains) at the putative epitopes of Nb H7 and H9, which reveal considerable variation. This is consistent with our ELISA results, showing only very weak binding of Nbs H7 and H9 to MeV H protein. We have consequently revised the Discussion (page 14, lines 455-459).

2. This study also lacks a comprehensive evaluation of the broad-neutralizing capabilities of Nbs and the synthetic antibody against the mutant CDV strains. Although the author claim that the epitopes of H7 and H9 are highly conserved among different CDV strains, many mutations (e.g. A544 site of the H7 epitope) were found in the epitopes of both antibodies. This raises concerns about the broad-spectrum activities of these antibodies. Consequently, the author should perform structural analysis or experiments to assess the impact of these mutations on both antibodies.

Authors: To assess the potential broad-spectrum activity of our nanobodies against circulating CDV strains, we performed quantitative membrane fusion assays (new Supplementary Table 7) rather than structural analyses. This approach allowed us to determine the potency of the nanobodies against the prototypic wild type H protein (strain A75/17) as well as against multiple, newly

designed H mutants. Selection of the mutations was performed based on amino acid residues mapping to the identified epitopes of Nb H7 and H9, and different in naturally circulating CDV strains (as well as the vaccine OL-CDV strain) (Supplementary Table 2 and 3).

Interestingly, our data clearly showed that the H mutation M500I conferred resistance to Nb H7, while the substitution G324W conferred resistance to Nb H9 (revised Supplementary Table 7). Because the mutation H-M500I is exclusively found in the vaccine strain (OL), this demonstrated the potential broad impact of Nb H7 among circulating CDV strains. Conversely, the G324W mutation is observed in the H protein of the vaccine strain (OL) as well as in the “South America 2” CDV lineage. Thus, Nb H9 potency may be reduced against certain CDV strains. Importantly, the biparatopic construct H7-H9 remained efficient against all tested H mutants.

Taken together, our results (revised Supplementary Table 7) provided evidence that the multivalent drug candidate may feature broad-spectrum activity against circulating CDV strains. A new paragraph describing these results has been added to the manuscript (pages 8-9; lines 261-285). We believe that this new dataset represents a valuable addition and further strengthens the revised manuscript.

3. This manuscript lacks a necessary discussion on the limitations of current CDV vaccines and does not provide a comprehensive review of existing research on CDV H protein-targeted drug, including studies on morbillivirus neutralizing antibodies. This omission conceals the need for developing antibody therapies. The authors should include a detailed summary of these topics in the Introduction section.

Authors: As suggested by the Reviewer, we included in the main text a sentence mentioning that distemper is still an important health issue in certain low-income countries, where vaccination of dogs remains unfortunately suboptimal (page 13, lines 439-440). Furthermore, and as mentioned in our initial submission, our drug has also a strong potential to provide clinical benefits in other specific cases (e.g., endangered animal populations in captive environments where the vaccine is not optimal).

Finally, as also suggested by the Reviewer, we added information about known inhibitors targeting the H protein of morbilliviruses (page 3, lines 93-95). Notably, beyond antibodies or soluble constructs of natural receptors, there is currently no additional strategy to block morbillivirus cell entry by targeting the H protein. Regarding inhibitors against the F protein, the necessary information can be found in the Introduction section (page 3; lines 85-90).

4. While Nb H7's receptor competition mechanism is well characterized, the precise mode of neutralization by Nb H9 remains speculative. The authors hypothesize that Nb H9 might disrupt H/F conformational coupling or F

triggering. However, no any evidence were provided. The author should provide more evidence (e.g. compare H-F complex stability with/without Nb H9) to addressing the mechanistic elucidation of Nb H9.

Authors: To address the mechanism of inhibition mediated by Nb H9, we conducted surface co-immunoprecipitation experiments. The goal was to investigate a potential interference on H/F complex stability mediated by Nb H9. The sensitivity of the assay, however, did not allow us to draw clear conclusions. Accordingly, the discussion about the mode of inhibition mediated by Nb H9 was kept as hypothesis in our revised manuscript (page 12; lines 396-399).

5. The risk of viral resistance is a critical concern in antibody drug development. In the animal assays conducted in this study, while most ferrets administered with multiple H7-H9-fFc injections were protected against a lethal CDV challenge. However, the delayed viral clearance and death of one (1/3) 4x-dose-treated ferret indicates that the potential emergence of resistant mutants. An S546Y mutant was indeed isolated from the administrated animal.

5.1. Although the authors demonstrated that the antibody could neutralize this mutant strain in vitro, however, this evidence alone did not support the conclusion that "No drug escape variant emerged upon treatment in vivo".

Authors: In accordance with the Reviewer's suggestion, we modified the subheading of this chapter by: "The H7-H9-fFc antibody neutralizes the CDV brain isolate" (page 11; line 355) as well as the last sentence of this chapter (page 11: lines 370-372).

5.2. Furthermore, the structural analysis of S546Y's impact on H7 binding appears unreasonable.

Authors: Extended Data Figure 8 has been removed from the manuscript.

5.3. The authors need to perform in vitro virus resistance selection experiments using H7, H9 and H7-H9-fFc respectively, to identify potential mutations impacting antibody binding, and compare escape mutation frequencies between dual-epitope and single-epitope antibodies.

Authors: While we agree that *in vitro* generation of escape variants and the determination of escape mutation frequencies between various Nb formats is an excellent strategy to discover mutations affecting the efficacy of monovalent and multivalent Nbs, we are convinced that the introduction of signature mutations found in selected circulating CDV strains and locating within the epitopes of Nb H7 and H9 represents an attractive alternative approach derived from real-world data. By conducting additional experiments based on the latter approach, we indeed discovered mutations that could interfere with Nb H7 (M500I) and H9 (G324W) efficacies. Importantly, when both nanobodies were combined in a

single construct (H7-H9), membrane fusion-inhibition was recorded against the single H mutants M500I and G324W as well as against the double H mutant (G324W/A325G). These results confirmed our hypothesis that combining two Nbs binding to spatially distinct epitopes and interfering with membrane fusion through different mechanisms of inhibition can better accommodate resistance mutations that are effective against their monovalent format. An additional paragraph describing those results was added in the revised manuscript (pages 8-9; lines 261-285) and the data are presented in a new Supplementary Table 7.

Finally, regarding the comment “the risk of viral resistance is a critical concern in antibody drug development”, we would like to emphasize that neutralizing immunity was elicited in all treated animals (including the one with the brain infection; Figure 5f). As only one serotype has been described so far for CDV and MeV (highlighting the lack of sufficient plasticity within the two envelope glycoproteins to efficiently escape binding of multiple neutralizing antibodies), such immunity is very likely protective against future infection with any CDV strains (including the brain isolate variant). Conversely, single serotype also suggests that vaccinated animals, or animals that survived a natural CDV infection, would be protected against the brain isolate. Thus, antibodies (and specifically nanobody-based multidomain Abs) as antiviral drugs to fight morbillivirus infections may translate into very promising drug candidates with potential real-world impact.

5.4. Additionally, the authors should also conduct an additional animal assay by challenging ferrets with the S546Y mutant to evaluate the efficacy of H7-H9-fFc.

Authors: While the suggested additional animal experiments with the brain isolate may be of interest, our aim is not generating drug escape variants through successive *in vivo* virus passaging. We therefore believe that such additional animal experiments lie beyond the scope of the present manuscript.

Minor comments:

1. The parameters for Poor Rotamer (1.37) and All-atom clash score (13.04) are relatively high, the structural model may require further refinement.

Authors: Based on the Reviewer's suggestion, we invested time and made efforts to improve the indicated parameters. As shown in the revised Supplementary Table 6, the values for poor/outlier rotamers and the all-atom clashscore could be improved.

2. The structural presentation in Figure 2 are confusing, please provide at least the entire cryo-EM density map color by elements and a dimeric model representation including one CDV-solH dimer bound by one H9 and two H7.

Authors: After discussion among the authors, we prefer to retain Figure 2 in its current form, as we consider it clear and easy to interpret, including for non-structural biologists. However, we have taken the Reviewer's feedback into account and have included a new Extended Data Figure 4 that provides the requested information and representation. We believe this addition represents a valuable enhancement to the revised version of our manuscript.

3. Clarify statistical methods, e.g., error bars in Fig. 5c,d; n-values for replicates.

Authors: A new paragraph specially describing the methods employed to generate statistics regarding the viral titers (Fig. 5d) was added in the Material and Method section (page 23, lines 785-788). We note that Fig. 5c has no error bars. The number of ferrets employed per condition was also added in the figure legend of Fig. 5 (page 39, line 1138) and Extended Data Fig. 8 (page 50, lines 1346 and 1351).

4. Please include the ethical clearance for this study.

Authors: We have added more details concerning ethical clearance in our revised manuscript (page 22, lines 757-759).

Additional corrections

We would like to apologize for an inaccurate data analysis regarding the presented K_D values and associated high standard deviations in our surface plasmon resonance studies. Upon careful re-analysis of the data, we now provide updated K_D values with accurate standard deviations. Note that the K_D values of both monovalent nanobodies remained very similar (Nb H7: 1.7 nM -> 2.3 nM; Nb H9: 3.8 nM -> 5.4 nM). Although the updated K_D value of the biparatopic is slightly decreased (0.2 nM -> 1.9 nM), it nevertheless remained the best binders to the CDV H protein (as presented in our initial analysis). Importantly, those slight differences in K_D values have no impact on any conclusions presented in our manuscript. Modifications were made in the main text (page 4, lines 125-127 and page 7, lines 234-236) as well as in the figure legends of the Extended Data Figure 1 (pages 41-42, lines 1168-1178). Table 1 was also modified accordingly (page 40) as well as panels b, c and e of Extended Figure 1 (page 41).

REVIEWERS' COMMENTS

Reviewer #1 (Remarks to the Author):

The authors have satisfactorily answered my critiques.

Reviewer #2 (Remarks to the Author):

The authors have addressed some of the previous concerns, however, several issues remain either unresolved or insufficiently addressed. Below are the major and minor points that still need attention.

Comment 1: In the rebuttal, although the authors provided additional ELISA data (not functional data), it remains insufficient as it does not fully address the core issue of cross-neutralization activity. The absence of neutralization assays against MeV leaves the key question unanswered, specifically whether the antibody has broader applicability to the Morbillivirus genus. Moreover, the weak binding efficiency to MeV H protein, as demonstrated in the additional ELISA, underscores that the study's scope is functionally limited to CDV, thereby reinforcing that the current title overstates the findings. While the CDV ferret model is well-established, it primarily validates an intervention against CDV itself. To legitimately claim genus-level protection, direct evidence, or at least cross-neutralization data against other morbilliviruses, particularly MeV, is essential, which the authors have not provided.

Authors: To avoid any confusions and overstatements regarding our findings, we modified the title by exchanging the word “morbillivirus” by “canine distemper virus” (page 1, line 1).

Comment 2: I commend the authors for utilizing quantitative membrane fusion assays to assess the broad-neutralizing capabilities of Nb H7, H9, and H7-H9 against mutant CDV strains. The data presented are compelling, demonstrating that the H7-H9 construct maintains efficacy against single mutants, thus strengthening the manuscript. However, at a concentration of 0.5 µg/ml, the dual mutant (G324W/A325G) significantly challenges the potency of H7-H9, suggesting reduced effectiveness. This finding implies that naturally occurring combinatorial mutations in circulating strains may compromise the H7-H9's broad-neutralizing capability. I recommend that the authors explicitly discuss this limitation in the manuscript to provide a comprehensive interpretation of the effectiveness of H7-H9.

Authors: We fully agree with the Reviewer's comment and clarified this point accordingly by adding sentences in the revised version of the manuscript (page 9-10, lines 297-300, and page 10, lines 305-307).

Comment 3: Addressed.

Comment 4: I acknowledge the authors' efforts in attempting co-immunoprecipitation experiments to elucidate the neutralization mechanism of Nb H9. However, as the assay yielded inconclusive results, the proposed mechanism remains speculative. The authors need to state in the text that the mechanism of Nb H9 neutralization remains undefined due to a lack of experimental evidence, ensuring conclusions are appropriately cautious.

Authors: As suggested by the reviewer, we modified two sentences in the revised version of our manuscript to make clear that the mechanism of neutralization mediated by Nb H9 remains undetermined (page 6, line 180, and page 13, lines 420-421).

Comment 5: In rebuttal, the authors demonstrated the use of quantitative membrane fusion assays to evaluate the broad-neutralizing capabilities of Nbs. However, the S546Y mutant, which was isolated from animal experiments and has significant practical implications, was not included in the in vitro evaluation. Considering that the authors did not intend to conduct the screening of escape mutants that I suggested, to thoroughly assess the antibody's drug resistance risk, it is recommended to include fusion inhibition assay data for the S546Y mutant. This inclusion would strengthen and substantiate the argument for the broad-neutralizing capability of H7-H9.

Authors: We employed our quantitative membrane fusion assay to collect the requested data regarding the H-protein mutant S546Y. Confirming our data obtained in our neutralization assay, membrane fusion activity triggered by the H-S546Y mutant could resist (to some extent) to Nb H7 but was completely blocked by Nb H9 as well as the biparatopic H7-H9 construct. Data are shown in an updated Supplementary Table 7, and a sentence has been added in the Result section (page 12, lines 392-393). Note that since additional values were inserted in the Supplementary Table 7, statistical analyses were also updated based on the entire dataset.

Minor comments: Addressed.